# Optimal enzyme utilization suggests that concentrations and thermodynamics determine binding mechanisms and enzyme saturations

Asli Sahin [1], Daniel R. Weilandt[1,2] & Vassily Hatzimanikatis [1] ✉

Deciphering the metabolic functions of organisms requires understanding the dynamic responses of living cells upon genetic and environmental perturbations, which in turn can be inferred from enzymatic activity. In this work, we investigate the optimal modes of operation for enzymes in terms of the evolutionary pressure driving them toward increased catalytic efficiency. We develop a framework using a mixed-integer formulation to assess the distribution of thermodynamic forces and enzyme states, providing detailed insights into the enzymatic mode of operation. We use this framework to explore Michaelis-Menten and random-ordered multi-substrate mechanisms. We show that optimal enzyme utilization is achieved by unique or alternative operating modes dependent on reactant concentrations. We find that in a bimolecular enzyme reaction, the random mechanism is optimal over any other ordered mechanism under physiological conditions. Our framework can investigate the optimal catalytic properties of complex enzyme mechanisms. It can further guide the directed evolution of enzymes and fill in the knowledge gaps in enzyme kinetics.

Living organisms constantly adapt to genetic or environmental perturbations. Describing these dynamic responses requires a deep understanding of the underlying network of biochemical and biophysical processes that comprise cellular metabolism. As cellular enzymes catalyze most metabolic processes, describing how perturbations propagate in large reaction networks requires knowledge and characterization of these enzymes' reaction mechanisms and kinetic properties. To this end, for over half a century, kinetic models of biochemical systems have been used to study a wide range of systems, from simple enzymatic reactions[1,2] and small pathways[3–6] to large-scale metabolic networks[7,8] with diverse applications in health, biotechnology, systems, and synthetic biology[9]. Over the last years, more efforts have been made toward building genome-scale kinetic models[10,11],

addressing and quantifying the uncertainty in the structure and parameters of kinetic models[12–15], and identifying metabolic engineering design strategies using kinetic models[16–18].

Kinetic models employ a mathematical description of the enzymatic reaction rates, defined precisely as a function of the metabolite concentrations and kinetic parameters. However, experimental data detailing the parameters of an enzyme are scarse[9,19]. Even large databases listing kinetic information, such as BRENDA[20] and SABIO-RK[21] do not comprise complete sets of parameters for the central metabolism of a single organism[19]. To address missing data, kinetic models use parameter estimation methods[11,22–25] or Monte Carlo sampling methods[8,13,15–17,26,27] that have proven useful. However, a complete understanding of the estimated parameters with biological and

[1]Laboratory of Computational Systems Biotechnology, Ecole Polytechnique Federale de Lausanne (EPFL), 1015 Lausanne, Switzerland. [2]Present address: Department of Chemistry and Lewis-Sigler Institute for Integrative Genomics, Princeton University, Princeton, NJ, USA. ✉e-mail: vassily.hatzimanikatis@epfl.ch

mechanistic details is generally not provided[28,29]. Nonetheless, unlike chemical systems, the parameters of biological systems are not random or unknowable but are, in fact, an outcome of natural selection, and they have evolved to fulfill their biological functions optimally[2,30–32]. The crucial point in the investigation of biological systems in the light of evolution is to formulate appropriate fitness functions whose maximum or minimum value potentially corresponds to an evolutionary outcome of the metabolism[28,33].

Various studies previously addressed the application of evolutionary principles to biological systems based on specific selective pressures. These studies range from characterizing isolated enzymes' kinetic responses[30,31,34] to analyzing the structure and function of metabolic networks[33], such as minimizing steady-state fluxes, transient times, and metabolic concentrations of intermediates, or maximizing the network thermodynamic efficiency[29]. These studies showed that exploring these parameters while accounting for the fact that they are an outcome of the evolutionary process can help us decipher the underlying design principles that govern enzyme catalytic rates.

One of the targets of natural selection on cellular metabolism is to make efficient use of its resources to grow, reproduce and respond to changes in their environments[28,35]. As cellular enzymes catalyze metabolic reactions, this selection will translate to evolutionary pressure toward optimal enzyme utilization. Therefore, the ratio of specific flux to enzyme concentration ($v_{net}/E_{tot}$) is an important determinant for the evolutionary optimization of cellular enzymes. This ratio depends on metabolite concentrations but also on the kinetic parameters of the enzyme. Over the last decades, with the advances in the genome-scale metabolic model reconstruction, more mathematical models have been developed to understand how evolution has shaped the system variables such as enzyme and metabolite concentrations[36–39] and kinetic parameters[30,34,40–42] and how these variables affect the network topology and the optimal utilization of enzyme. These modeling approaches differ in assumptions made, mechanistic details included, and evolutionary objectives studied. Furthermore, these studies employ different mathematical optimization methods ranging from deterministic methods, including convex[36,38] and non-convex[30,34,37] problems, to stochastic methods with population-based algorithms[40–42]. (see Supplementary Note for further details).

It has often been stated that evolutionary pressure drives enzymes toward maximal catalytic efficiency such that enzyme utilization is optimized. The hypothesis is strongly supported by the high reaction rates observed for the enzyme-catalyzed reactions compared to their corresponding uncatalyzed reactions[29]. One of the early examples of a catalytically efficient enzyme is the triosephosphate isomerase (TIM/TPI) shown by Knowles and Albery[32]. Although recent meta-studies analyzing a large dataset of available enzyme kinetic parameters suggest that the evolution drives most enzymes toward "good enough" rather than perfect[41,43], we still have limited information on the driving forces and the constraints that have shaped natural enzymes. Understanding the fitness landscape of enzymes toward catalytic optimality can improve our understanding of the parameters that govern the design of enzymes and potentially overcome the scarcity of kinetic parameters.

Previous studies have addressed the hypothesis of catalytic optimality either by employing a population-based optimization method[40–42] or by solving a nonlinear optimization problem[30,34]. The existing population-based approaches do not account for the reaction kinetics in detail. Instead, they focus on maximal catalytic rates[40] without detailed modeling of the enzyme kinetics and thermodynamics terms[40,41] or on simplified reaction mechanisms[42]. These population-based approaches, furthermore, rely on extensive hyperparameters optimization to perform adequately and cannot ensure global optimality or convergence. Unlike the existing population-based approaches, Heinrich and coworkers investigated the catalytic optimality of unbranched enzymatic mechanisms with detailed

reaction rate equations by solving a nonlinear optimization problem[30,34]. These studies investigated the kinetic parameters of ordered enzyme mechanisms at enzyme constrained maximal catalytic activity. Their results indicated that reactant concentrations significantly impacted the optimal rate constants, dividing the concentration space into different sub-regions, with distinct binding characteristics[28,30,34]. They also have shown that the reactant concentrations and Michaelis constants change in the same direction in an evolutionary time-scale[30,34]. Their findings are corroborated by experimental observations[44,45]. Although these studies were useful for understanding enzyme evolution, their approach is limited to ordered enzyme mechanisms and cannot account for alternative enzyme mechanisms such as Ping-Pong or random mechanisms[46]. Furthermore, their approach relied on an initial step where they first derived all possible types of optimal solutions and then solved the nonlinear problem locally for each optimal solution with the Lagrange multipliers method[30,34]. As cells contain hundreds to thousands of enzymatic reactions with different mechanisms, deriving exact solutions for numerous mechanisms can be cumbersome and, in some cases, not possible with the existing formulation. Hence it is necessary to develop efficient computational frameworks to explore the space of catalytic efficiencies for arbitrarily complex reaction mechanisms.

In this study, we have developed a computationally efficient mixed-integer linear program (MILP) formulation. Our framework, which we named OpEn (OPtimal ENzyme) estimates optimal kinetic parameters of complex enzyme mechanisms and assesses the coupling between thermodynamic displacements, saturation, and elementary rate constants at the optimal state. The presented framework provides insights into the selective pressures that shape the catalytic efficiency of enzymes. Furthermore, it can be used to estimate parameters for kinetic models, filling in the knowledge gaps in enzyme kinetics from an evolutionary perspective and providing a method for improving the accuracy of metabolic models.

## Results

### A generalized framework to study optimal enzyme utilization for arbitrary elementary mechanisms

In the presented work, we study how enzymatic reactions operate if the total amount of enzyme is utilized optimally under the biophysical constraints posed by nature. We, therefore, use an optimization formulation to maximize the net steady-state flux given a fixed amount of enzyme level, as has been done in previous studies[30,34,47]. We wanted to ensure that the OpEn framework applied to all enzymatic mechanisms with known elementary reaction schemes and could directly assess the distribution of thermodynamic forces and enzyme states, providing detailed insight into the mode of operation. Therefore, we formulated our optimization problem using as inputs: (i) the elementary enzyme mechanism, (ii) the intracellular concentrations of the substrates and products, and (iii) their thermodynamic properties in terms of the standard Gibbs free energy of the reactions (Fig. 1 panel Inputs). The operating conditions that our framework examines and provides as outputs comprise (i) a set of elementary rate constants, (ii) elementary thermodynamic displacements (i.e., equivalent to the thermodynamic driving forces), and (iii) the distribution of the enzyme states (i.e., the relative allocation of the total amount of enzyme to substrate-bound, product-bound, or free states) (Fig. 1 panel 4).

To achieve this goal, we formulated four sets of biophysical constraints within our framework. First, we assumed that the enzyme operates at a quasi-steady state. Thus, the concentrations of substrates, product, and enzyme states are time-invariant, resulting in a set of equality constraints. Secondly, we set the total amount of enzyme as constant by assuming that its transcription and translation dynamics are sufficiently slow compared to its metabolic dynamics. Furthermore, we linked the ratio of the elementary forward and reverse fluxes to their respective thermodynamic force $\gamma_i$[48]. Finally, we

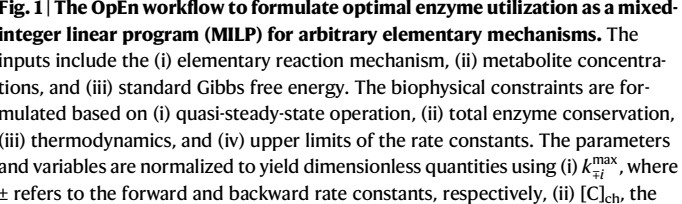

**Fig. 1 | The OpEn workflow to formulate optimal enzyme utilization as a mixed-integer linear program (MILP) for arbitrary elementary mechanisms.** The inputs include the (i) elementary reaction mechanism, (ii) metabolite concentrations, and (iii) standard Gibbs free energy. The biophysical constraints are formulated based on (i) quasi-steady-state operation, (ii) total enzyme conservation, (iii) thermodynamics, and (iv) upper limits of the rate constants. The parameters and variables are normalized to yield dimensionless quantities using (i) $k_{\mp i}^{\max}$, where $\pm$ refers to the forward and backward rate constants, respectively, (ii) $[C]_{ch}$, the characteristic concentration, and (iii) $[E_T]$, total enzyme concentration. The constraints are then linearized to overcome the non-linearity of the problem by applying a (i) change of variables and (ii) piecewise-constant approximation of the independent displacement variables. The enzyme utilization is then optimized using the MILP formulation by maximizing the net steady-state flux of the enzymatic reaction. Here, analyses can be performed on the mode of operation at optimality.

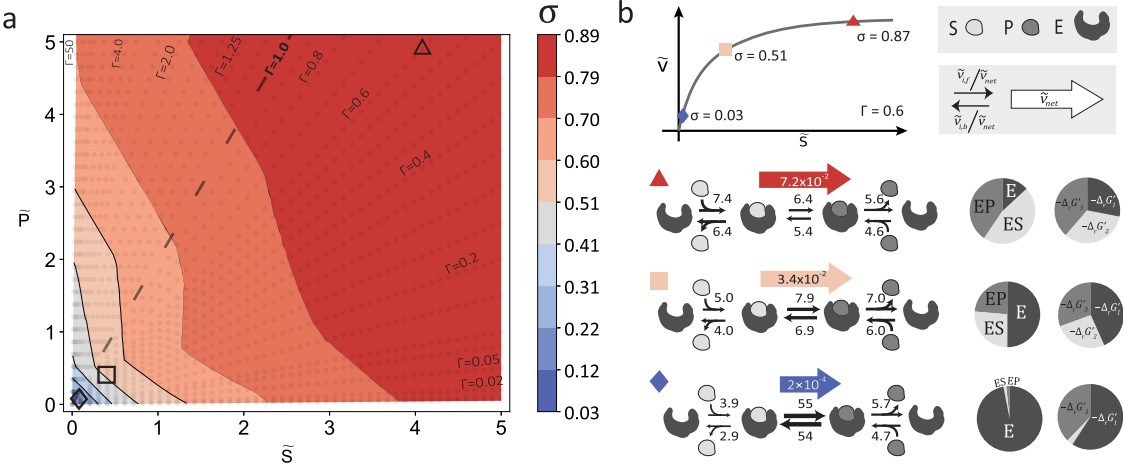

**Fig. 2 | Modes of operations of the optimally used Michaelis-Menten enzyme.**
**a** Enzyme saturation (σ) of the reversible Michaelis–Menten mechanism (see Eq. (1)) at a catalytically optimal state. Scattered isolines indicate the displacements from thermodynamic equilibrium (Γ's). Dashed line indicates the equilibrium (Γ = 1.0). **b** Three different operating points along the Γ = 0.6 isoline with low (blue), medium (nude), and high (red) saturations and their respective operating conditions for optimal enzyme utilization, net-flux, elementary fluxes, enzyme state, and free energy distribution, for $\widetilde{K}_{eq} = 2$ (data for different $\widetilde{K}_{eq}$'s can be found in Supplementary Fig. 4). Subscripts in the pie chart refer to the elementary steps denoted in Eq. (1). Source data are provided as a Source Data file.

considered biophysical limits[33,49] for the elementary rate constants by limiting bimolecular rate constants by their diffusion limit, varying within the range $10^8 - 10^{10} M^{-1} s^{-1}$[33,41]. The monomolecular rate constants are limited by the frequency of molecular vibrations, which was found to vary in the interval $10^4 - 10^6 s^{-1}$ for enzymatic reactions[34,47] (Fig. 1 panel 1).

Designed to mimic considerations that would apply in cells, the formulation of the biophysical constraints encompasses four sets of variables and two sets of parameters for a given enzyme mechanism. The variables consist of the elementary rate constants $k_{\mp i}(k_{i,f}, k_{i,b})$, thermodynamic displacements $\gamma_i$, and enzyme states $e_i$, and the parameters consist of the metabolite concentrations and the overall equilibrium constant $K_{eq}$ (or the overall thermodynamic displacement Γ).

To obtain dimensionless quantities, we normalize certain variables and parameters, namely rate constants $k_{i,f}$ and $k_{i,b}$, enzyme states $e_i$, metabolite concentrations [P] [S], and the overall equilibrium constant $K_{eq}$. We normalize the elementary rate constants by their respective biophysical limits as done previously by Wilhelm et al.[34]. To normalize metabolite concentrations and the overall equilibrium constant, we also used the aforementioned limits and introduced a characteristic concentration $[C]_{ch}$. Lastly, we normalized the enzyme states using the total enzyme concentration. This yields $\widetilde{S}, \widetilde{P}, \widetilde{K}_{eq}$ as parameters and $\widetilde{k}_{i,f}, \widetilde{k}_{i,b}, \widetilde{e}_i$ and $\gamma_i$ as variables (see Methods and Fig. 1 panel 2).

To overcome the nonlinear nature of the problem, we next linearized the bilinear terms and the nonlinear constraints. As the normalized elementary rate constants $\widetilde{k}_{\mp i}(\widetilde{k}_{i,f}, \widetilde{k}_{i,b})$ are well-bounded between 0 and 1, we replaced the bilinear terms $\widetilde{k}_{\mp i}\widetilde{e}_i$ for each elementary step by one new variable and one new constraint. To replace the nonlinear constraint posed by the thermodynamics, we eliminated one of the displacements using the overall thermodynamic displacement from equilibrium. We estimate the remaining elementary displacements or mechanistically meaningful independent combinations of them by a piecewise-constant function. The resulting problem is piecewise-linear and can be solved efficiently with a MILP formulation using the Peterson linearization scheme[50,51] (See Methods). The reformulation of the problem as a MILP ensures global optimality and enumeration of alternative solutions (Fig. 1 panel 3).

Finally, the resulting MILP allows us to optimize the net enzyme flux for the respective operating conditions, i.e., substrate, product

concentrations, and standard Gibbs free energy. The optimization results will yield a set of elementary rate constants, displacements, and an enzyme distribution that allow for this optimal flux. With this constraint-based formulation of the problem, we can then assess potential alternative modes of operation by constraining the flux to its maximum and applying variability analysis on the operational variables. With this same principle, we can study the fitness landscape of optimal enzyme utilization for specific operating conditions by exploring the suboptimal space (Fig. 1 panel 4).

## Optimally used Michaelis–Menten enzymes require condition-specific saturation regimes

We first applied our framework to study the modes of operation of the prototypical three-step reversible Michaelis–Menten Eq. (1) mechanism. Our results from our herein presented elementary MILP formulation capture those previously obtained by Wilhelm et al, we showed the division of the concentration space into distinct regions, with matching elementary rate constants and Michaelis constants at optimal state[34] (Supplementary Figs. 1 and 2). We further calculated forward and backward turnover numbers at optimal state and showed their sensitivity over the concentration space (Supplementary Fig. 3).

$$E + S \underset{k_{1,b}}{\overset{k_{1,f}}{\rightleftharpoons}} ES \underset{k_{2,b}}{\overset{k_{2,f}}{\rightleftharpoons}} EP \underset{k_{3,b}}{\overset{k_{3,f}}{\rightleftharpoons}} E + P \qquad (1)$$

Additionally, our formulation allowed us to assess optimal enzyme state distributions and thermodynamic forces directly. Our results show that the operating conditions govern the enzyme state and thermodynamic force distribution at a catalytically optimal state.

A comprehensive analysis of enzyme-state distributions showed that optimal enzyme utilization requires the enzyme to operate at a low enzyme saturation if the substrate and product concentrations are small compared to the characteristic concentration of the system. With increasing substrate and product concentrations, the optimal enzyme utilization requires increasing enzyme saturation. The optimal saturation increases rapidly with substrate and product concentration when both reactant concentrations are below the characteristic concentration $[C]_{ch}$, whereas for larger substrate or product concentrations, this increase is significantly smaller (Fig. 2a). To our surprise, this

phenomenon appears to be independent of the thermodynamic displacement Γ (Fig. 2a).

To better understand the precise mechanism by which this strong dependence of the optimal saturations emerges, we analyzed the characteristic operating conditions for the low, intermediate and high saturation regimes (Fig. 2b). The data revealed three optimal prototypical mechanisms by which the optimal enzyme utilization is achieved:

At low saturations, the thermodynamic driving force (ΔG′) of the reaction was mainly used for substrate association, which required about 60% of the potential. Most of the remaining thermodynamic potential is used to drive the product dissociation, and only a minimal amount is allocated to displace the biotransformation step from equilibrium. This distribution of the thermodynamic forces manifests itself in a fast turnover between the enzyme-bound substrate and product and a comparatively slow turnover for substrate and product association and dissociation. As indicated by the low saturation, most enzymes are free enzymes, providing the necessary driving force to capture the substrate molecules present in low quantities.

At intermediate saturations, the thermodynamic driving force (ΔG′) of the biotransformation step increased to the same order of magnitude as the product dissociation. This shift in thermodynamic forces came mainly at the expense of the driving force for the substrate association reaction, with some contribution from reducing the driving force of the product dissociation reaction. This redistribution resulted in an overall reduced contribution of the substrate association and product binding steps compared to the low saturation case, with the substrate association being the slowest step. Most enzymes remain free indicating that capturing the substrate molecules is still a limiting factor.

At high saturations, the thermodynamic driving forces are equally distributed, resulting in a similar turnover between the three steps. Comparing the actual elementary fluxes shows that the product dissociation association has become the slowest step after the biotransformation and the substrate association dissociation exhibited the fastest turnover. Furthermore, free enzyme became the least abundant species, showing that with increasing substrate concentrations, the capture of substrate molecules becomes less of a limiting factor. Nevertheless, the decreasing dependency of the enzyme saturation on product and substrate concentration also indicates the minimum amount of free enzyme is required for the enzyme to operate optimally.

**Optimality principles of multi-substrate enzymes indicate concentrations dependent binding preferences**

To next examine more complex mechanisms, we applied our framework to investigate the optimal modes of operation of multi-substrate enzymes. To this end, we first studied the Bi-Uni mechanism (2) with a compulsory order for substrate binding. Our MILP formulation can capture subdivision of the concentration space into different regions classified based on the rate-constants, which was previously derived by Wilhelm et al.[34]. (Supplementary Fig. 5)

$$E + A \underset{k_{1,b}}{\overset{k_{1,f}}{\rightleftharpoons}} EA + B \underset{k_{2,b}}{\overset{k_{2,f}}{\rightleftharpoons}} EAB \underset{k_{3,b}}{\overset{k_{3,f}}{\rightleftharpoons}} EP \underset{k_{4,b}}{\overset{k_{4,f}}{\rightleftharpoons}} E + P \qquad (2)$$

In addition to the elementary rate constants, we again captured enzyme saturation at different operating conditions for the optimally utilized enzyme. The analysis of the enzyme-state distributions revealed that similar to the reversible three-step Michaelis-Menten mechanism, saturation at optimal state increases with product concentration (see Supplementary Fig. 6). Interestingly, we observed that the order in which the substrates bind to the enzyme plays a role in saturation at a catalytically optimal state, with saturation increasing with an increased concentration of the substrate that binds first. Our

results show that saturation increases with the increasing substrate concentration that binds first to the enzyme. In contrast, the concentration of the second substrate does not significantly change the saturation.

Since our generalized MILP formulation allows us to study any kind of elementary mechanisms in an unbiased fashion, we extended the scope to investigate a generalized Bi-Uni mechanism, where any substrate can bind first to the enzyme, resulting in the following branched mechanism (see Eq. (3)).

$$
\begin{array}{c}
E + A \underset{k_{1,b}}{\overset{k_{1,f}}{\rightleftharpoons}} EA \\
+ \qquad\qquad + \\
B \qquad\qquad B \\
k_{5,f}\updownarrow k_{5,b} \qquad k_{2,b}\updownarrow k_{2,f} \\
EB + A \underset{k_{6,b}}{\overset{k_{6,f}}{\rightleftharpoons}} EAB \underset{k_{3,b}}{\overset{k_{3,f}}{\rightleftharpoons}} EP \underset{k_{4,b}}{\overset{k_{4,f}}{\rightleftharpoons}} E + P
\end{array} \qquad (3)
$$

Our results suggest that optimal enzyme have a preferential binding mechanism dependent on their operating conditions. To quantify this preference, we introduced the splitting ratio, $\alpha = \tilde{v}_{net,up}/\tilde{v}_{net}$, which is defined as the fraction of the flux that goes through the upper branch, where substrate A binds first to the enzyme (see Eq. (3)).

Interestingly, the values of optimal splitting ratios are found to vary between 0.3 and 0.7 under physiological conditions ($\tilde{P} = 1 \approx 0.1\,mM$), indicating that the random-ordered mechanism is optimal over ordered mechanism (Fig. 3a). A detailed analysis of the optimal splitting ratio across various input scenarios revealed three phenomenological features for the substrate binding preference. First, if the substrate concentrations are interchanged symmetrically ($\tilde{A}, \tilde{B} \to \tilde{B}, \tilde{A}$), the optimal splitting ratio $\alpha$ switches, leaving $\alpha_{B,A} = 1 - \alpha_{A,B}$ (Fig. 3a–c). This demonstrates that the splitting ratio shows an antisymmetric behavior with symmetric changes in the substrate concentrations.

Secondly, we observed that for distinct concentrations of the substrates ($\tilde{A} \neq \tilde{B}$), the splitting ratio and the optimal modes of operation were unique, though when both substrates were equally available ($\tilde{A} = \tilde{B}$), the splitting ratio behaved flexibly. A variability analysis revealed that this flexibility had a range around $\alpha = 0.5$. This led to alternative values for the elementary rate constants and, consequently to alternative modes of net fluxes through the upper and lower branches. More interestingly, when the substrate concentrations are equal and less than product concentrations, there exists a unique configuration of elementary constants for the optimally utilized enzymes; instead, when they are comparable or slightly higher than product concentrations, there can exist alternative values for the elementary constants that can result in optimal enzyme utilization. This suggests that the product concentration relative to the concentration of the substrates affects the uniqueness of the optimal solution and therefore, the preferential binding for the unbiased Bi-Uni mechanism.

Lastly, our findings revealed a relationship between the preferential binding mechanism and the reactant concentrations. As mentioned above, the preferential binding mechanism shows an antisymmetric behavior over the substrate concentration space. However, when one substrate concentration is greater than the other ($\tilde{B} > \tilde{A}$ or $\tilde{A} > \tilde{B}$), such as at the upper or lower concentration spaces, the data reveals two characteristic behaviors for the preference of binding to either the lower or higher abundant substrate, as demonstrated by the net flux through the upper or lower pathway. (i) When the concentration of the least abundant substrate (e.g. $\tilde{B}$ for the lower and $\tilde{A}$ for the upper concentration space) is lower than the product concentration $\tilde{P}$, the most abundant substrate binds first to the

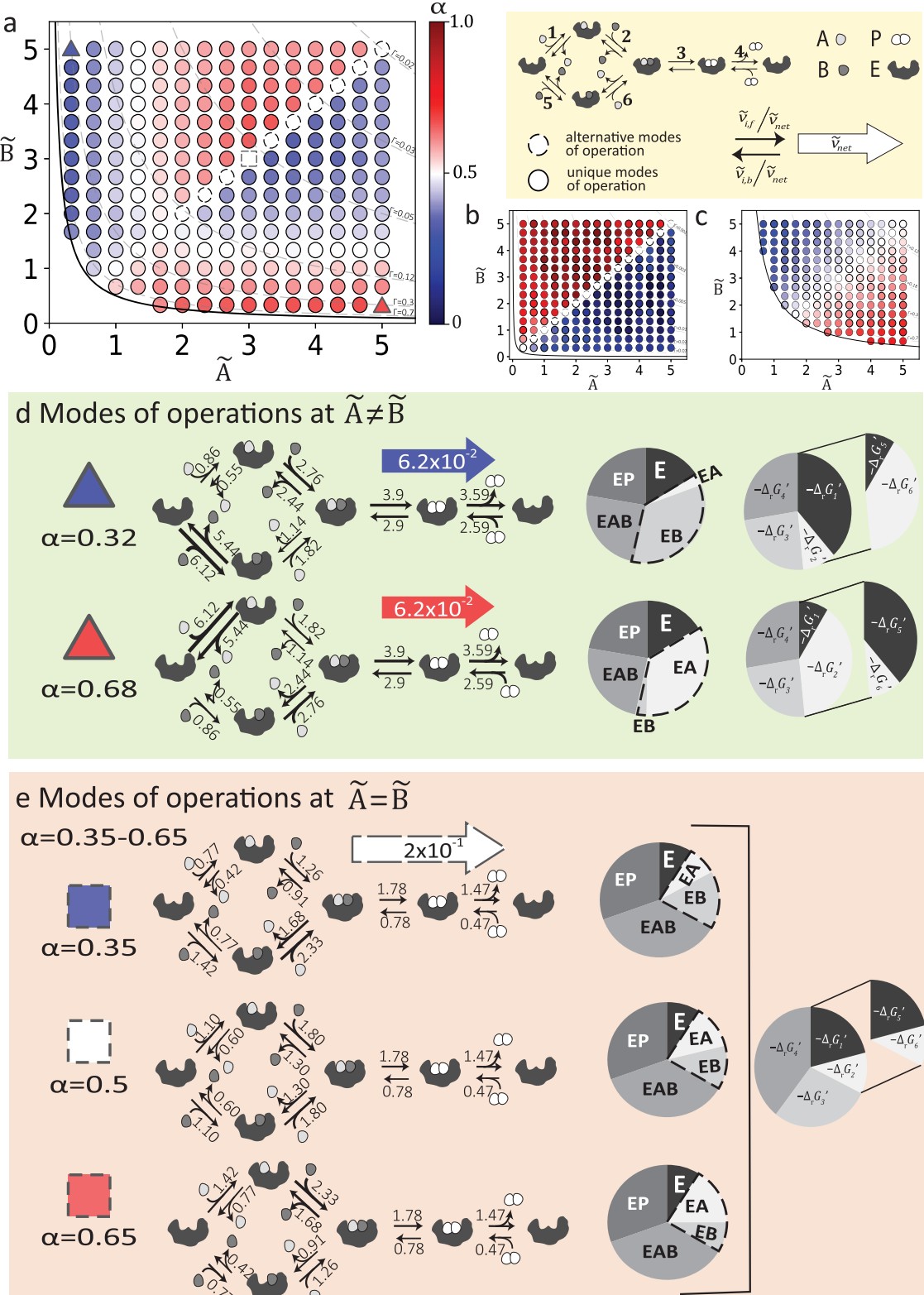

enzyme. (ii) On the contrary, when the concentration of the least abundant substrate is comparable to or higher than the concentration of the product, the least abundant substrate binds first to the enzyme.

It was surprising that the substrate concentrations were not the sole determinants for these behaviors (Fig. 3a), and that the product concentration played such a determining role. Looking into this further, the shift between the two behaviors can be seen clearly when the product concentration is equal to 1 ($\tilde{P}=1$) (Fig. 3a). At low product

concentrations, such as $\tilde{P}=0.1$, the lowest substrate concentration defines the preference for the binding (Fig. 3b). Likewise, at high product concentrations ($\tilde{P}=5$), the most abundant substrate concentration defines the preferential binding mechanism (Fig. 3c). Our results suggest that contrary to common belief, the preferential binding mechanism is not only determined by the substrate concentrations but also by their magnitudes in relation to the product concentration.

**Fig. 3 | Optimal splitting ratio $\alpha = \frac{\tilde{v}_{net,up}}{\tilde{v}_{net}}$ and modes of operations for the general bi-uni mechanism on the concentration space of the substrates $\tilde{A}$ and $\tilde{B}$.** Optimal splitting ratio for the general bi-uni mechanism for $\tilde{K}_{eq} = 2$. **a** for $\tilde{P} = 1$ **b** for $\tilde{P} = 0.1$ and **c** for $\tilde{P} = 5$, colors indicate the magnitude of $\alpha$, if $\alpha > 0.5$ (red) most of the flux at optimal state goes through the branch where A binds first to the enzyme, and similarly if $\alpha < 0.5$ (blue) most of the flux goes through the branch where B binds first. A splitting ratio of 0.5 (white) indicates that the flux is equally distributed between both branches. Dashed line style for the scatters indicates the flexibility of the splitting ratio at optimal state. Solid line indicates the equilibrium, dashed isolines indicate the displacements from equilibrium. Prototypical operating

conditions for optimal enzyme utilization, net-flux, elementary fluxes, enzyme state, and free energy distribution, for the selected data points indicated in part a, for $\tilde{P} = 1$ (triangles and a square) **d** for distinct ($\tilde{A} \neq \tilde{B}$), and symmetric concentrations of subtrates with unique modes of operations (triangles), **e** when both substrates are equally available ($\tilde{A} = \tilde{B}$) (square). Subscripts in the pie chart refer to the elementary steps denoted in Eq. (3). Free energy pie charts have an alternate segment representing the upper and lower branches of the reaction mechanism. Dashed lines in the enzyme state pie charts indicate the substrate bound enzyme states, and its conservation across symmetric and alternative solutions. Source data are provided as a Source Data file.

To decipher the mechanistic details behind these phenomena, we analyzed the modes of operation at an optimal state. Similar to the reversible Michaelis-Menten mechanism, we analyzed three prototypical operating conditions: two symmetric operating conditions for distinct substrate concentrations (red and blue triangles in Fig. 3a) and one with identical concentrations for the substrates (white dashed square in Fig. 3a). Here, the symmetric substrate concentrations resulted in symmetric modes of operation. In other words, there was a symmetrical interchange between the upper and lower branches for the thermodynamic forces, net fluxes, and enzyme-state distributions. Otherwise, the modes of operation for the biotransformation and the product association-dissociation steps remain identical (Fig. 3d, e).

For the selected symmetrical operating conditions, preferential binding went to the most abundant substrate. Therefore, when the concentration of B was high, 68% of the net flux went through the lower branch, resulting in $\alpha = 0.32$ (Fig. 3d). Here, around 40% of the thermodynamic potential drove the association-dissociation steps for substrate A (steps 1 or 6), which can be attributed to the low concentration of A at this condition. Most of the remaining potential drove the biotransformation and the product association-dissociation steps, whereas a minimal amount of potential was allocated to the binding of substrate B (steps 2 or 5). The highest turnover was observed for the association-dissociation step of substrate B to the free enzyme, which was then followed by the biotransformation and product association-dissociation steps. The slowest turnover was observed for the association and dissociation steps for substrate A (steps 1 or 6).

A similar analysis also applies for the symmetric operating condition in which the concentration of A is high. In this case, 68% of the net flux goes through the upper branch, resulting in $\alpha = 0.68$ (Fig. 3e). Nearly 40% of the thermodynamic potential drove the association-dissociation reactions for substrate B, followed by the biotransformation and the product association-dissociation steps. The lowest amount of potential was used to drive the association-dissociation of substrate A.

As a result of the symmetric modes of operations, identical optimal saturation is observed for symmetric operating conditions. Further analysis of the enzyme-state distributions revealed that the total concentration allocated to the substrate-bound and product-bound enzyme states stays the same, leaving the saturation unchanged between symmetric operating conditions. The only difference arises for the specific substrate-bound enzyme states, namely EA and EB, which interchange for symmetric operating conditions. If most of the flux goes through the branch where the substrate B binds first to the enzyme, B-bound enzyme state, EB, will be more occupied than the A-bound enzyme state, EA (Fig. 3d). Likewise, for the symmetric operating condition, EA concentration will increase at the expense of reducing the amount attributed to the EB concentration (Fig. 3d).

A detailed analysis of the modes of operation at flexible operating points shows that the saturation and the thermodynamic force displacement remain the same across the alternative solutions (Fig. 3e). Nevertheless, the flexibility in splitting ratio expresses itself as alternating flux distributions for the branched pathway and on the distribution of the substrate-bound enzyme states, EA and EB. The

occupancy of the remaining enzyme states and the flux distributions through the biocatalysis and product dissociation steps remain the same for all alternative solutions, indicating that the saturation at optimal state remains unique across alternative solutions.

## Discussion

In this study, we presented a computational method to explore the catalytically optimal modes of operations of enzymatic reactions as a framework to eventually help fill in missing enzyme kinetic information in metabolic models. We formulated a MILP problem maximizing the net-steady state reaction rate at a given total enzyme concentration, and estimated the optimal kinetic parameters, saturation and thermodynamic force displacements for the three-step reversible Michaelis–Menten mechanism and, for the first time, a random-ordered multi-substrate mechanism. The results from our framework matched with optimized rate constants from previous studies[34,47] and further expanded the field to provide details about the condition-specific saturations and binding mechanisms that are required for optimal enzyme utilization.

Our formulation has three clear advantages over the existing methods focusing on the catalytic optimality of enzymes. First, the formulation allows us to address any complex enzyme mechanism, such as random-ordered multi-substrate mechanisms. Second, by combining our formulation with the traditional sampling methods, we can explore the suboptimal solutions and study the fitness landscape of optimal enzyme utilization. Lastly, the MILP formulation ensures global optimality and convergence, which can be solved by commercial solvers, thus we do not need to derive solutions for all possible types of kinetic designs, as was done in previous studies[34,47]. Instead, we showed the emergence of diverse optimal kinetic designs by solving the optimization problem at given reactant concentrations and thermodynamic constraints. We expect future work to be extended to account for different concentrations and constraints as needed, such as for studying the influence of biochemical reactions occurring within densely packed cells, where enzymes operate in highly crowded environments which dramatically alters the Michaelis–Menten parameters from dilute solutions[52].

An added value of our framework is the limited necessary inputs which are the reaction mechanism, thermodynamic properties of the reaction, and the metabolite concentrations. The reaction mechanism can be assigned based on the molecularity of the reaction or on previous knowledge and literature[9]. The standard Gibbs free energies of the reactions can be estimated by group-contribution[53] or component-contribution[54] methods. Lastly, the metabolite concentrations can be obtained partially from the literature for well-studied organisms[44,45], or can be estimated for an entire metabolic pathway using constraint-based optimization methods such as Thermodynamics-based Flux Analysis (TFA)[55,56]. Using TFA[55,56], we can integrate quantitative metabolomics and fluxomics data and estimate thermodynamically feasible concentration and flux profiles as well as how far each reaction operates from thermodynamic equilibrium by sampling the thermodynamically feasible concentration and flux profile space, as it is done in the construction of population of kinetics models[13]. After having the

necessary inputs, our framework can be easily applied to study a network of enzymatic reactions or an entire metabolic pathway under the assumption that each reaction operates at a catalytically optimal state. This procedure scales linearly with the number of reactions in the network. Thus, by combining experimental measurements with genome-scale metabolic models and constraint-based modeling approaches, our framework can estimate a context-specific theoretical upper bound for the catalytic efficiency of metabolic enzymes with detailed kinetic and thermodynamic considerations as they operate in vivo.

In this study, our analysis was focused on and limited to the kinetic design and modes of operations of enzymes at maximal reaction rates, though our computational framework would also allow the exploration of alternative and suboptimal solutions. Therefore, although we are still far from completely understanding the complex interplay between the physicochemical constraints and evolutionary pressures that shape enzyme catalysis, our framework can help us map the parametric domain for a broad range of operating conditions and shed light on the driving forces and constraints that have shaped natural enzymes in three ways. First of all, using our framework, we can estimate a condition-specific theoretical upper bound for the catalytic efficiency of any enzyme mechanism, which will provide a more accurate comparison with the natural enzymes to assess how far they operate from their theoretical optimum. Secondly, as we formulate the problem as a MILP, we can explore the suboptimal solutions and explore the fitness landscape of enzymes toward catalytic perfection, which was not possible with previous methodologies. This way, we can understand the independent contribution of each variable to the fitness landscape of the moderately efficient enzymes and have a detailed understanding of how diminishing returns and trade-offs affect the evolutionary trajectory of enzymes toward optimal catalytic efficiency. Lastly, we can integrate alternative objectives by including them in the objective function to study different evolutionary pressures and trade-offs between them.

Overall, we expect the presented framework to be used in the future to estimate the missing kinetic parameters for steady-state flux profiles in metabolic models, thereby overcoming the scarcity of kinetic parameters and advancing the development of kinetic models. The use of our presented framework will allow some of the first studies into complex enzyme mechanisms in the light of evolution, which will directly fill knowledge gaps in enzyme kinetics. Additionally, the provided estimated modes of operation at a catalytically optimal state can be translated into enzyme bioengineering strategies, guiding the design of enzymes for maximal catalytic activity for direct application to common engineering problems, such as large-scale product syntheses and therapeutic development.

## Methods
### Normalization of parameters
For our formulation, we used the normalized parameters and variables as was done in previous studies[34,47]. Additionally, we also normalized the concentrations of the enzyme states in terms of the total enzyme concentration (see (Fig. 1) panel 2). We considered two limits for the elementary rate-constants, one for the bimolecular (second-order) rate constants, $k_{\pm i}^{\max,2}$ and one for the monomolecular (first-order) rate constants, $k_{\pm i}^{\max,1}$. We did not distinguish between the isomerization and dissociation steps and constrained both rate constants with the same upper limit. Nevertheless, the presented methodology can be generalized by introducing different upper limits for different types of monomolecular rate constants, as was done previously by Klipp and Heinrich[30,47].

## Describing the rate of reaction
We describe the reaction rate at the elementary reaction level using mass-action kinetics. Considering that all the elementary steps are reversible, at steady state, the reaction rates are written by decomposing each reversible flux into two separate irreversible fluxes. To do this, we introduced displacements from thermodynamic equilibrium $\gamma_i$ for each elementary step, for $i = 1, \ldots, N_e$ where $N_e$ is the number of elementary reactions.

$$\gamma_i = \frac{v_{i,b}}{v_{i,f}} \tag{4}$$

For convenience throughout this section, instead of $+$ ,-. the subscripts $f$ and $b$ are used to denote forward and backward variables, where $v_{i,f}$ and $v_{i,b}$ denotes the forward and backward reaction rates, respectively for the $i^{th}$ elementary step. We assume that the net reaction rate is positive, that is the reaction operates in the forward direction. This implies that $\gamma_i \leq 1$ for $i \in \{1, \ldots, N_e\}$.

The displacement of a reaction from its thermodynamic equilibrium $\Gamma$, is defined as the ratio between the backward reaction rate $\widetilde{v}_b$ to the forward reaction rate, $\widetilde{v}_f$ for the overall reaction and is defined as follows for a reaction with n substrates and m products:[48,57]

$$\Gamma = \frac{\widetilde{v}_b}{\widetilde{v}_f} = \frac{1}{\widetilde{K}_{eq}} \frac{\prod_{k=1}^{m} \widetilde{P}_k}{\prod_{j=1}^{n} \widetilde{S}_j}. \tag{5}$$

$$\widetilde{v}_f - \widetilde{v}_b = \widetilde{v}_{net}. \tag{6}$$

Here, $\widetilde{K}_{eq}$ stands for the reaction equilibrium constant and is defined as $\frac{\prod_{k=1}^{m} \bar{P}_k^{eq}}{\prod_{j=1}^{n} \bar{S}_j^{eq}} = e^{-\triangle_r G^{\circ}/RT}$, where $\triangle_r G^{\circ}$ is the standard Gibbs free energy of the reaction; R is the ideal gas constant and T is the temperature. $\widetilde{v}_{net}$ is the net steady-state flux for the overall reaction and is defined as the difference between the forward and backward reaction rates. The Gibbs free energy of the reaction can be written as:

$$\triangle_r G' = \triangle_r G^{\circ} + RT \ln \frac{\prod_{k=1}^{m} \widetilde{P}_k}{\prod_{j=1}^{n} \widetilde{S}_j}. \tag{7}$$

Using Eqs. (4) and (7), $\Gamma$ can also be expressed as:

$$\Gamma = e^{\triangle_r G'/RT}. \tag{8}$$

Consequently, for reactions operating in the direction of product production, the Gibbs free energy of the reaction is negative and $\Gamma \in [0,1]$. Similarly, if the reaction operates in the reverse direction, $\Delta_r G'$ is positive and $\Gamma \in [1, +\infty]$. Note that $\Gamma$ close to 1 indicates a reaction operating close to equilibrium.

$\Gamma$ is linked to the elementary displacements according to the following equation:

$$\Gamma = \prod_{k \in C} \gamma_k. \tag{9}$$

The multiplication is over a set C, and its content depends on the kinetic mechanism of the reaction. For unbranched enzymatic reactions (e.g., ordered mechanisms), set C contains all elementary reactions. For a random-ordered mechanism, Eq. (9) needs to be satisfied

for each fundamental cycle[58], $\forall C \in C_f$, where set $C$ is a subset of $C_f$, which contains all fundamental cycles.

As we assume that the reaction proceeds toward the production of products, Eq. (9) constraints the thermodynamic displacements of elementary reactions as follows: $\Gamma \leq \gamma_i \leq 1$. Without loss-of-generality the presented framework can also be applied for the reactions operating in the reverse direction (toward substrate production) by applying: $\Gamma \to \frac{1}{\Gamma}$, $\gamma_i \to \frac{1}{\gamma_i}$, $\widetilde{K}_{eq} \to \frac{1}{K_{eq}}$ and $\widetilde{v}_{net} \to -\widetilde{v}_{net}$

The net steady-state reaction rate can be written from $2N_e$ equality constraints, where $N_e$ is the number of elementary steps.

$$\widetilde{v}_{i,f} = \widetilde{k}_{i,f} \widetilde{e}_{i-1} \widetilde{c}_i = \frac{\widetilde{v}_{i,net}}{(1-\gamma_i)}. \tag{10}$$

$$\widetilde{v}_{i,b} = \widetilde{k}_{i,b} \widetilde{e}_i \widetilde{c}_i = \frac{\widetilde{v}_{i,net}\gamma_i}{(1-\gamma_i)}. \tag{11}$$

$$\widetilde{v}_{i,net} - \widetilde{k}_{i,f} \widetilde{e}_{i-1} \widetilde{c}_i (1-\gamma_i) = 0. \tag{12}$$

$$\widetilde{v}_{i,net}\gamma_i - \widetilde{k}_{i,b} \widetilde{e}_i \widetilde{c}_i (1-\gamma_i) = 0. \tag{13}$$

Here, $\widetilde{v}_{net,i}$ denotes the net steady-state flux for the elementary reaction i, $\widetilde{k}_{i,f}$ and $\widetilde{k}_{i,b}$ stand for the forward and backward elementary rate constants of the i$^{th}$ elementary step ($i \in \{1,\ldots,N_e\}$), $\widetilde{e}_i$ is the corresponding abundance of the enzyme state for the i$^{th}$ elementary step, and $\widetilde{e}_0 = \widetilde{e}_n$. The cyclic notation holds for ordered enzyme mechanisms, whereas for random-ordered mechanisms the corresponding enzyme state can be generated using the King-Altman method[59]. Furthermore, $\widetilde{c}_i$ is the concentration of the reactants, which is a parameter in our formulation, and is equal to 1 for all dissociation steps and interconversion steps or is equal to the concentration of the substrates or the products for the $i$th association step, for substrate or product binding steps.

Note that the net steady-state fluxes for elementary reactions are the same as the net flux for the overall reaction, for unbranched mechanisms $\widetilde{v}_{i,net} = \widetilde{v}_{net}$. For random-ordered mechanisms (see Eq. (3)), the following relation must be added: $\widetilde{v}_{j,net} = \widetilde{v}_{net} j \in M$, $\sum_{r \in B_k} \widetilde{v}_{r,net} = \widetilde{v}_{net}$. The set M contains all the elementary steps in the unbranched pathway and the set B contains all combinations for the elementary steps ($B_k$) from each branch in the mechanism (for Eq. (3) $B \in [\{1,5\},\{2,5\},\{1,6\},\{2,6\}]$). Using the ratio $\widetilde{v}_{r,net}/\widetilde{v}_{net}$ we also define the splitting ratio $\alpha$ for the random-ordered mechanisms.

Considering the conservation of total enzyme adds an additional constraint:

$$\sum_{n=1}^{N} \widetilde{e}_n = 1. \tag{14}$$

The sum is over all enzyme mechanistic states for a given mechanism, where N is the total number of enzyme states. $N = N_e$ for ordered mechanisms, and $N = N_e - n_b$ for random-ordered mechanisms, where $n_b$ is the number of branching points in a mechanism. As enzyme states are normalized with the total enzyme concentration, the right side of Eq. (14) is equal to 1.

With all the constraints described our optimization problem can be stated as follows:

$$
\begin{aligned}
\max \quad & \widetilde{v}_{net} \\
\text{s.t} \quad & & \forall i = 1, \cdots, N_e \\
& \widetilde{v}_{i,net} - \widetilde{k}_{i,f} \widetilde{e}_{i-1} \widetilde{c}_i (1-\gamma_i) = 0, \\
& \widetilde{v}_{i,net}\gamma_i - \widetilde{k}_{i,b} \widetilde{e}_i \widetilde{c}_i (1-\gamma_i) = 0, & \forall i = 1, \cdots, N_e \\
& \widetilde{v}_{j,net} - \widetilde{v}_{net} = 0, & \forall j \in M \\
& \sum_{r \in B_k} \widetilde{v}_{r,net} - \widetilde{v}_{net} = 0, & \forall B_k \in B \\
& \prod_{k \in C} \gamma_k = \Gamma, & \forall C \in C_f \\
& \sum_{n=1}^{N} \widetilde{e}_n = 1, \\
& \widetilde{k}_{i,f} \leq 1, \widetilde{k}_{i,b} \leq 1, & \forall i = 1, \cdots, N_e \\
& \Gamma \leq \gamma_i \leq 1, & \forall i = 1, \cdots, N_e \\
& \widetilde{v}_{i,net}, \widetilde{k}_{i,f}, \widetilde{k}_{i,b}, \widetilde{e}_i, \gamma_i \in \mathbb{R}_+. & \forall i = 1, \cdots, N_e
\end{aligned}
\tag{15}
$$

### Change of variables
Due to the non-linearity of the rate equation given by Eqs. (12) and (13), we first apply the change of variables. We replaced each of the bilinear terms denoting the multiplication of elementary rate constants and the corresponding enzyme state variables by a new variable ($\widetilde{z}_{i,f}$ and $\widetilde{z}_{i,b}$) and a constraint. As described above, elementary rate constants are normalized with their corresponding biophysical limit; hence they can take values in the interval [0,1]. Thus, replacing the bilinear terms with new variables implies that they ($\widetilde{z}_{i,f}$ and $\widetilde{z}_{i,b}$) are bounded above by their corresponding enzyme states. Reformulating of Eqs. (12) and (13) with the change of variables results in the following constraints:

$$\widetilde{v}_{i,net} - \widetilde{z}_{i,f} \widetilde{c}_i (1-\gamma_i) = 0. \tag{16}$$

$$\widetilde{v}_{i,net}\gamma_i - \widetilde{z}_{i,b} \widetilde{c}_i (1-\gamma_i) = 0. \tag{17}$$

$$\widetilde{z}_{i,f} - \widetilde{e}_{i-1} \leq 0. \tag{18}$$

$$\widetilde{z}_{i,b} - \widetilde{e}_i \leq 0. \tag{19}$$

Introducing new variables removes elementary rate constants from the rate equation, leaving the $\widetilde{z}$, $\widetilde{e}$ $\gamma$'s and $\widetilde{v}_{net}$ as variables of the optimization problem.

### Approximation of the elementary displacements from thermodynamic equilibrium
To remove the remaining non-linearity in Eqs. (16) and (17), we approximated the elementary displacements from thermodynamic equilibrium ($\gamma_i$'s) with a piecewise-constant function $\hat{\gamma}_i$, or in other words, with a 0$^{th}$-order approximation. If $\hat{\gamma}_i$ is piecewise-constant, then the products $\widetilde{v}_{i,net}\hat{\gamma}_i$, $\widetilde{z}_{i,f}\hat{\gamma}_i$, and $\widetilde{z}_{i,b}\hat{\gamma}_i$ are piecewise-linear and can be described in MILP form. This approximation converts the continuous bilinear terms into mixed bilinear terms, which are a product of an integer and a continuous term. This simplified the problem, as these mixed bilinear terms can be linearized in an MILP formulation using the Petersen linearization scheme[50,51], which was previously used for metabolic engineering[60,61].

Approximations of the thermodynamic displacements also needed to satisfy the overall thermodynamic constraint stated in Eq. (9). As we have an explicit definition of elementary displacements in the

rate equations, Eq. (9) also accounts for the ratio of the elementary rate constants that satisfies the overall equilibrium constant. First, we eliminated one of the elementary displacements using the overall thermodynamic displacement ($\Gamma$). Then we approximated the independent elementary displacements or their mechanistically meaningful combinations with a piecewise-constant function.

As we are interested in reactions operating toward the production of products: $\gamma_i \in [\Gamma,1]$. Then we approximated $\gamma_i$ using the following equation:

$$\gamma_k \approx \hat{\gamma}_k = \Gamma + p\frac{(1-\Gamma)}{N}, \qquad (20)$$

where, $\frac{(1-\Gamma)}{N}$ is the resolution of the approximation, $N$ is the number of bins into which $\hat{\gamma}_i$ has been discretized, and p chooses which bin is selected for the solution. Here $k \in I_c$ and set $I_c$ denotes the chosen elementary displacement variables or their combinations to be approximated with a 0th order approximation. To linearize the problem, we expressed $p$ using binary variables. For this, we represent $p$ with its binary expansion.

$$p = \sum_{s=0}^{\lceil \log_2 N \rceil} 2^s \delta_s, \qquad (21)$$

where, $\lceil \log_2 N \rceil$ indicates the smallest majoring integer to $\log_2 N$, and $\delta_s$ is the binary variable $\delta_s \in \{0,1\}$. As we performed binary expansion of p, the complexity of the model increased with $\mathcal{O}(\log_2 N)$ instead of $\mathcal{O}(N)$, which was also previously used by Salvy and Hatzimanikatis[61]. We also needed to ensure that p does not exceed N:

$$0 \le \sum_{s=0}^{\lceil \log_2 N \rceil} 2^s \delta_s \le N. \qquad (22)$$

For simplicity in our formulation, we approximated meaningful combinations of elementary displacements to reduce the number of linearization to be performed. For example, for the reversible Michaelis–Menten reaction given in Eq. (1), we chose two independent displacement variables to linearize as $\hat{\gamma}_1$ and $\hat{\gamma}_{1,2}$, where the latter represents the product of $\hat{\gamma}_1$ and $\hat{\gamma}_2$ ($\hat{\gamma}_{1,2} = \hat{\gamma}_1\hat{\gamma}_2$), which also adds the constraint $\hat{\gamma}_{1,2} - \hat{\gamma}_1 \le 0$. In this way, we could represent each elementary displacement as $\gamma_1 \approx \hat{\gamma}_1$, $\gamma_2 \approx \frac{\hat{\gamma}_{1,2}}{\hat{\gamma}_1}$, and $\gamma_3 \approx \frac{\Gamma}{\hat{\gamma}_{1,2}}$. Any other combination of elementary displacements from equilibrium works for this mechanism. Overall, we performed the piecewise-constant approximation for the chosen independent elementary displacement variables $\gamma_k \approx \hat{\gamma}_k$, $k \in I_c$.

The approximation became more important for the combination of elementary displacements when we studied random-ordered mechanisms. For the random-ordered mechanism given in Eq. (3), by approximating the combination of elementary displacements for the cycle e.g $\hat{\gamma}_{cycle} = \hat{\gamma}_{1,2} = \hat{\gamma}_{5,6}$, we could directly satisfy Eq. (9) for each fundamental cycle describing the principle of microscopic reversibility[58]. Thus, for the random-ordered Bi-Uni mechanism, we could express all six elementary displacements ($\gamma_i$) by approximating four independent displacement variables, namely: $\hat{\gamma}_{cycle},\hat{\gamma}_1,\hat{\gamma}_5,\hat{\gamma}_3$ ($I_c$ = cycle,1,5,3), which yields $\gamma_1 \approx \hat{\gamma}_1$, $\gamma_5 \approx \hat{\gamma}_5$, $\gamma_2 \approx \hat{\gamma}_{cycle}/\hat{\gamma}_1$, $\gamma_6 \approx \hat{\gamma}_{cycle}/\hat{\gamma}_5$, $\gamma_3 \approx \hat{\gamma}_3$, and $\gamma_4 \approx \Gamma/(\hat{\gamma}_{cycle}\hat{\gamma}_3)$. Note that the constraints $\hat{\gamma}_{cycle} - \hat{\gamma}_1 \le 0$ and $\hat{\gamma}_{cycle} - \hat{\gamma}_5 \le 0$ also need to be considered.

For the results, the resolution of the approximation $\frac{1-\Gamma}{N}$ of elementary displacements was set to $10^{-4}$ for the Michaelis–Menten and $10^{-3}$ for the random-ordered Bi-Uni mechanisms. If the reaction operates close-to equilibrium, $\Gamma \ge 0.9$, $\frac{1-\Gamma}{N} = 10^{-4}$ for both mechanisms.

## Petersen linearization

After approximating elementary displacements from equilibrium, the rate equation contained the bilinear terms arising from the products $\tilde{v}_{i,net}\hat{\gamma}_i$, $\tilde{z}_{i,f}\hat{\gamma}_i$, and $\tilde{z}_{i,b}\hat{\gamma}_i$. We could approximate these continuous products using the following derivation, which is only shown for the product $\tilde{v}_{net,i}\hat{\gamma}_i$ for simplicity. The same linearization scheme also applies to the remaining nonlinearities of the form $\tilde{z}_{i,fwd}\hat{\gamma}_i$ and $\tilde{z}_{i,bwd}\hat{\gamma}_i$.

$$\tilde{v}_{i,net}\gamma_i \approx \tilde{v}_{i,net}\hat{\gamma}_i. \qquad (23)$$

$$\tilde{v}_{i,net}\hat{\gamma}_i = \tilde{v}_{i,net}\Gamma + \sum_{s=0}^{\lceil \log_2 N \rceil} \frac{(1-\Gamma)}{N}2^s \delta_s \tilde{v}_{i,net} \qquad (24)$$

The product $\delta_s \tilde{v}_{i,net}$ is bilinear, though is a product of a binary and a continuous variable. Assuming a constant $M > \tilde{v}_{i,net}$, we could apply the Petersen linearization scheme[50,51] to the bilinearity. Replacing $\delta_s \tilde{v}_{i,net}$ with another non-negative variable $t_s^i$, where s stands for the index of the binary variable and i stands for the elementary step, we could represent the bilinear product by one new variable and three new constraints:

$$t_s^i = \delta_s \tilde{v}_{i,net}. \qquad (25)$$

$$\begin{cases} \tilde{v}_{net,i} + M\delta_s - t_s^i \le M \\ t_s^i - M\delta_s \le 0 \\ t_s^i - \tilde{v}_{i,net} \le 0 \end{cases} \qquad (26)$$

Note that when $N_e > 3$, we need an additional linearization to account for the product of two binary variables. As an example, consider the random-ordered Bi-Uni mechanism given in Eq. (3). By approximating 4 elementary displacement variables ($\hat{\gamma}_{cycle},\hat{\gamma}_1,\hat{\gamma}_5,\hat{\gamma}_3$) we can express all 6 elementary displacements as explained in the previous section. To describe the reaction rate from the product dissociation step, we can use the following equations:

$$\tilde{v}_{4,net} - \tilde{z}_{4,f}(1-\gamma_4) = 0 \qquad (27)$$

$$\tilde{v}_{4,net}\gamma_4 - \tilde{z}_{4,b}(1-\gamma_4)[\tilde{P}] = 0 \qquad (28)$$

$$\tilde{z}_{4,f} - [\overline{EP}] \le 0 \qquad (29)$$

$$\tilde{z}_{4,b} - [\tilde{E}] \le 0 \qquad (30)$$

Note that the product-dissociation step $\tilde{v}_{4,net} = \tilde{v}_{net}$ is not in the branched pathway of the reaction mechanism. We then needed to represent $\gamma_4$ from the approximated elementary displacements as $\hat{\gamma}_4 = \Gamma/(\hat{\gamma}_{cycle}\hat{\gamma}_3)$.

We can rewrite the constraints above using the following equations:

$$\tilde{v}_{net}\hat{\gamma}_{cycle}\hat{\gamma}_3 - \tilde{z}_{4,f}(\hat{\gamma}_{cycle}\hat{\gamma}_3 - \Gamma) = 0 \qquad (31)$$

$$\tilde{v}_{net}\Gamma - \tilde{z}_{4,b}(\hat{\gamma}_{cycle}\hat{\gamma}_3 - \Gamma)[\tilde{P}] = 0 \qquad (32)$$

$$\tilde{z}_{4,f} - [\overline{EP}] \le 0 \qquad (33)$$

$$\tilde{z}_{4,b} - [\tilde{E}] \le 0 \qquad (34)$$

Here $\hat{\gamma}_{cycle}$ and $\hat{\gamma}_3$ are the approximations by a piecewise-constant function as described by Eqs. (20)–(24). As both approximations contain binary variables, their product needed to be considered. This product could be linearized by representing it with a new binary variable and three new constraints as follows:

$$\zeta_{s,p} = \delta_s \lambda_p \tag{35}$$

$$\begin{cases} \zeta_{s,p} - \delta_s \le 0 \\ \zeta_{s,p} - \lambda_p \le 0 \\ \zeta_{s,p} - \delta_s - \lambda_p + 1 \ge 0 \end{cases} \tag{36}$$

where $\zeta_{s,p}$, $\delta_s$ and $\lambda_p$ are binary variables, $\zeta_{s,p}, \delta_s, \lambda_p \in \{0,1\}$. $\delta_s$ and $\lambda_p$ are the binary variables in the binary expansion for $\hat{\gamma}_{cycle}$ and $\hat{\gamma}_3$, respectively. After this linearization, the remaining bilinearity was of the form *continuous × binary*, which could be linearized using Petersen's theorem[50,51] (Eqs. (25)–(26)).

Using the change of variables and the piecewise-constant approximation as described above, we translated the nonlinear optimization problem given in Eq. (15) to a MILP, which can be summarized as follows:

$$\begin{aligned}
\max \quad & \tilde{v}_{net} \\
\text{s.t} \quad & \\
& \left.\begin{aligned} \tilde{v}_{i,net} - \tilde{z}_{i,f}\tilde{c}_i(1-\hat{\gamma}_i) &= 0 \\ \tilde{v}_{i,net}\hat{\gamma}_i - \tilde{z}_{i,b}\tilde{c}_i(1-\hat{\gamma}_i) &= 0 \\ \tilde{z}_{i,f} - \tilde{e}_{i-1} &\le 0 \\ \tilde{z}_{i,b} - \tilde{e}_i &\le 0 \end{aligned}\right\} & \forall i = 1, \cdots, N_e \\
& \tilde{v}_{j,net} - \tilde{v}_{net} = 0 & \forall j \in M \\
& \sum_{r \in B_k} \tilde{v}_{r,net} - \tilde{v}_{net} = 0 & \forall B_k \in B \\
& \hat{\gamma}_k = \Gamma + \sum_{s=0}^{\log_2 N} \frac{(1-\Gamma)}{N} 2^{s,k} \delta_{s,k} & \forall k \in I_C \\
& \sum_{n=1}^{N} \tilde{e}_n = 1 \\
& \Gamma \le \hat{\gamma}_i \le 1 & \forall i = 1, \cdots, N_e \\
& \tilde{v}_{i,net}, \tilde{z}_{i,f}, \tilde{z}_{i,b}, \tilde{e}_i, \hat{\gamma}_i \in \mathbb{R}_+ & \forall i = 1, \cdots, N_e \\
& \delta_{s,k} \cdot \in \{0,1\} & \forall k \in I_C
\end{aligned} \tag{37}$$

Note that, the overall thermodynamic constraint was dropped in the final formulation, as approximations of the independent elementary displacement variables ($\hat{\gamma}_k$) were performed accordingly as explained before.

### Variability analysis

The optimality in MILPs ensures that there is a unique global optimum value for the objective function ($\tilde{v}^*_{net}$) but not a unique optimal value for the variables. Therefore, there can be multiplicity of solutions. To account for this multiplicity, variability analysis was performed for the variables of the problem by finding the maximum and minimum values of each variable at a given state (e.g. optimal state).

### Back calculation of elementary rate-constants

Our MILP formulation does not consider the elementary rate constants $\tilde{k}_{i,f}$ and $\tilde{k}_{i,b}$ as explicit variables of the optimization problem. Instead, they are embedded in the linearized variables $\tilde{z}_{i,f}$ and $\tilde{z}_{i,b}$, the product of the elementary rate constants with their corresponding enzyme states. To back calculate the elementary-rate constants, we performed a variability analysis for the variables $\tilde{z}_{i,f}$, $\tilde{z}_{i,b}$, and $\tilde{e}_i$, where $i \in \{1, \ldots, N_e\}$ at the optimal state ($\tilde{v}_{net} = \tilde{v}^*_{net}$). For the ordered mechanisms, we observed that the optimal state is achieved by unique values for the $\tilde{z}$, $\tilde{e}$, and $\gamma$'s. This implies that the values for the elementary rate

constants are also unique and can be calculated by dividing $\tilde{z}_{i,f}$ and $\tilde{z}_{i,b}$ by $\tilde{e}_{i-1}$ and $\tilde{e}_i$, respectively. The uniqueness of the solution for the ordered mechanisms was also previously shown by Wilhelm et al.[34].

For the random-ordered mechanisms, the maximal catalytic activity was achieved by unique or alternative solutions depending on the reactant concentrations. First, we performed variability analysis on the steady-state fluxes of the branched elementary steps and calculated the flexibility of the splitting ratio ($\alpha = v_{r,net}/v_{net}$). For the flexible operating points, elementary displacements were uniformly sampled within their allowed range (calculated with variability analysis) and their values ($\gamma_i's$) were fixed for each feasible thermodynamic displacement distribution. Then the model became solely linear, and we sampled the variables with traditional sampling techniques, such as artificially centered hit and run (ACHR)[62] or optGpSampler[63]. After sampling, we could back calculate the elementary rate constants and study alternative modes of operation at the given reactant concentrations (Fig. 3).

### Sampling suboptimal solutions

In this study, we focused on the modes of operation of enzymes achieving maximal net steady-state flux (at the optimal state). Alternatively, as we formulated the problem as a MILP, we could also study suboptimal solutions that are at or beyond a given cut-off value with the constraint: $\tilde{v}_{net} \ge c_l \tilde{v}^*_{net}$, where $c_l$ denotes the cut-off limit. Thereby, using a similar procedure as in the previous section, we could explore the suboptimal space with sampling. In this way, we can explore the modes of operation of "moderately efficient" enzymes and study their fitness landscape toward catalytic perfection.

### Calculation of macroscopic kinetic parameters

We described the reaction rates from their elementary reaction mechanisms and estimated the corresponding microscopic rate constants for each elementary step. Translating the microscopic parameters to macroscopic ones ($K_M$'s and $k_{cat}$'s) can be performed using Cleland's notation[64] (see Supplementary Information) or equivalently by performing in silico initial rate experiments[52]. In this way, the estimated macroscopic parameters at the optimal state can be directly compared with available experimental data.

### Reporting summary

Further information on research design is available in the Nature Portfolio Reporting Summary linked to this article.

## Data availability

Source data are provided with this paper for all the figures generated in this study in the Supplementary Information/Source Data file. All datasets generated during this study can also be found in the repository https://github.com/EPFL-LCSB/open under the data subfolder. Source data are provided with this paper.

## Code availability

The implementation of this framework was performed in Python 3.6 using the optlang package[65] and using commercial solvers such as ILOG CPLEX or Gurobi. Code was run in Docker (20.10.6) containers. OptGP and ACHR samplers are adapted from their implementation in COBRApy[66] version 0.17.1. The code to use the workflow and to reproduce the results presented herein are available in the repository https://github.com/EPFL-LCSB/open. The code is also deposited in Zenodo to provide a reference to the version used in this study[67].

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

## Acknowledgements

The authors would like to thank to Dr. Kaycie Butler for her valuable input on the wording and structure of this paper. Funding for this work was provided by Swiss National Science Foundation (SNSF): grant 200021_188623 (A.S.), NCCR Microbiomes, a National Center of Competence in Research, funded by SNSF grant 51NF40_180575 (D.R.W.), the European Union's Horizon 2020 Research and Innovation Programme under grant agreements No. 686070 and 814408 (D.R.W), and the École Polytechnique Fédérale de Lausanne.

## Author contributions

A.S., D.R.W., and V.H. conceptualized the study. A.S. developed the software and performed the simulations. A.S., D.R.W., and V.H. analyzed the results and provided the discussion. A.S., D.R.W., and V.H wrote and reviewed the manuscript. D.R.W and V.H. managed and supervised the project. V.H. acquired the funding and the resources.

## Competing interests

The authors declare no competing interest.
