## [Peer review file · Nature Communications]

REVIEWER COMMENTS

Reviewer #1 (Remarks to the Author):

This manuscript reports the development of an MILP method to investigate the optimal modes of enzyme operation under the premise of evolutionary pressure to maximise catalytic efficiency. The method is validated by demonstrating it can reproduce results generated by previous authors for the simpler cases of a Michaelis-Menten single substrate enzyme and a compulsory order two substrate mechanism. However, this new algorithm is able to find the optimum states for more complex enzyme mechanisms such as the common two substrate random-order mechanism.

The key new result is the demonstration that the random order mechanism results in a more efficient optimum than the compulsory order mechanism. As the authors point out, the optimisation method would allow inference of plausible parameter sets for enzymes for which (as is usual) complete experimental sets are lacking, thus aiding the construction of more realistic and large scale kinetic metabolic models.

The methodology is fully described, and the software is made available for other researchers.

My comments relate to relatively minor points.

1. The specific references cited in para 1, l 23, give the impression that only the Hatzimanikatis and Maranas groups have contributed to such the development and application of kinetic metabolic models. Whilst an exhaustive review is not appropriate, it would be fairer to give some indication that such endeavours have been under way for over half a century since David Garfinkel, for example.

2. l 34, first sentence. This is an extreme understatement. Readers would be given a better idea of the scale of the problem by pointing out that even large databases of kinetic parameters such as BRENDA cannot furnish complete sets of parameters for central metabolism of any organism.

3. l.194/5: "substrate product association dissociation"? Was "substrate enzyme" intended?

4. Fig. 3. Needs a magnification of about 2X to become decipherable, especially as there is a lot of fine detail to be absorbed. It would have helped if the legend had pointed out that the free energy pie charts

had alternate segments representing the two branches of the mechanism rather than leaving the reader to decipher it.

5. Ref. 21: "Hermann-Georg" is the Holzhütter of ref 25.

6. Supplementary material has only two references, but the citation numbers are not always 1 and 2.

Reviewer #2 (Remarks to the Author):

The authors present a framework to find optimal values for the kinetic constants in an enzymatic mechanism using MILP. This article follows in the tradition of the seminal works by Knowles and others and it is specially relevant now that optimization frameworks have proven to be valuable tools for analyzing other aspects of biochemistry and metabolism such as stoichiometry and thermodynamics.

The application of the framework produces results in agreement with the existing literature for simple mechanisms and enables the analysis of virtually any mechanism.

Right now, techniques like Flux Balance Analysis and its thermodynamical extensions enable an understanding of physiologically meaningful metabolic states in terms of flux distributions. Simultaneously, recent breakthroughs are showing the logic behind the allocation of resources in the cells with strategies such as proteome fractioning. The missing link between these two approaches is the catalytic efficiency of enzymes, a topic that is still poorly understood from a theoretical perspective. This article provides a valuable tool to help understand the evolution of enzymatic mechanisms.

The manuscript is well written, the methods are sound and the authors provide the code necessary for reproducing the results. I recommend accepting this manuscript as it is.

Reviewer #3 (Remarks to the Author):

Sahin et al submitted the manuscript entitled "Optimal enzyme utilization suggests that concentrations and thermodynamics determine the operating binding mechanisms and enzyme saturations". The paper presents a theoretical work and an associated implemented MILP framework aiming at predicting the thermodynamics and kinetic parameters of Michaelis-Menten enzymatic systems. The originality of the work lies in the limited input data required for the prediction - elementary enzyme mechanisms, concentrations of substrates and products, thermodynamics properties - and the application to a random-ordered multi-substrate system. The manuscript is very well written, and demonstrates the quality of the work. The code associated to the manuscript works and is well documented.

My comments are the following:

How could this theoretical work be applied in practice to networks of biochemical reactions? If one wants to analyse a full metabolic pathway (several enzymes working as a cascade) or a larger network, is it possible and how would they proceed?

An added value of the work is the limited necessary inputs. How easy are they to obtain in practice and how does it prevent the scalability of the predictions to large(r) enzymatic systems? You mention in the introduction the difficulty for a cell containing thousands of enzymatic reactions, but not how your approach actually facilitates the task.

Could you provide validation of your optimisation approach on existing experimental data?

Results L216-224: is all the paragraph about FigS6? If yes, maybe refer to it earlier (eg L217-219)

Results on multi-substrate enzymes.

- I could not relate the observations to the figure at L246-250

- L273-274, can you refer to the blue and red triangles of Fig 3a and white triangle of fig 3a?

- L285 "...binding of substrate B": maybe add "(steps 2 or 5)"?

Discussion L344: any possible explanation on why enzymes do not operate at optimal efficiency and how your model could help understand the reasons?

Figure 1. The figure is great but it has a lot of contents associated in the main text and it is hard to see which part of the figure the text refers to. Could you refer to the specific subparts of the figure in the text (e.g. Figure 1 panel 1) rather than just adding "Figure 1"?

Figure 2. Panel a. Could you add information about the thermodynamic variable Γ in the legend? And mention as well the dashed line for $\Gamma = 1$?

Figure 3.

- Panel d has a lot of content and is referred to quite a few times in the text. Would it make sense to split d into d and e, d being the part of the blue and red triangles?
- The text in the pie plots is a bit small
- The legend could be extended a bit: what do the dotted lines in the pie charts mean? Why are the right pie charts combined to a sub-pie chart?
- The 3 alpha values in the white triangles (0.35 to 0.65) are unclear at first sight as they correspond to a single triangle in 3a. Maybe emphasize on the meaning in the legend of lower d, or instead you could choose a color different from white in a) for those points (because white is 0.5).

Supplementary figures:

- the right subpanels in several supp figs are not self-contained as the y-axis label is only displayed on the left subpanel: could you add the label to the other plots?
- Fig S2. A dashed line is mentioned in Fig S2's legend but is not present in the plot.
- Fig S3. I couldn't find any reference to it in the text.

Ref 34 of main text and ref 2 of supp show a switch between the first name and name of the first author.

Authors' response to the reviewers' comments

Title: Optimal enzyme utilization suggests that concentrations and thermodynamics determine the operating binding mechanisms and enzyme saturations

Manuscript ID: NCOMMS-22-42599

Authors: Sahin A., Weilandt D. R., and Hatzimanikatis V.

General comment:

We thank the reviewers for their positive feedback and insightful and constructive comments and suggestions that have helped us improve the manuscript's quality and clarity. The responses to the reviewers' comments and suggestions, proposed improvements, and modifications are highlighted in blue below. In addition, we have highlighted in blue the modifications done in the manuscript to include the reviewer's suggestions.

We hope the reviewers find that the revised manuscript addresses their concerns.

Reviewer comments

Reviewer #1 (Remarks to the Author):

This manuscript reports the development of an MILP method to investigate the optimal modes of enzyme operation under the premise of evolutionary pressure to maximise catalytic efficiency. The method is validated by demonstrating it can reproduce results generated by previous authors for the simpler cases of a Michaelis-Menten single substrate enzyme and a compulsory order two substrate mechanism. However, this new algorithm is able to find the optimum states for more complex enzyme mechanisms such as the common two substrate random-order mechanism.

The key new result is the demonstration that the random order mechanism results in a more efficient optimum than the compulsory order mechanism. As the authors point out, the optimisation method would allow inference of plausible parameter sets for enzymes for which (as is usual) complete experimental sets are lacking, thus aiding the construction of more realistic and large scale kinetic metabolic models.

The methodology is fully described, and the software is made available for other researchers.

We thank the reviewer for the positive and encouraging words.

My comments relate to relatively minor points.

1. The specific references cited in para 1, l 23, give the impression that only the Hatzimanikatis and Maranas groups have contributed to such the development and application of kinetic metabolic models. Whilst an exhaustive review is not appropriate, it would be fairer to give some indication that such endeavours have been under way for over half a century since David Garfinkel, for example.

We thank the reviewer for his remark, and we included the following references in the introduction:

- Chance, B., Garfinkel, D., Higgins, J., Hess, B. & Chance, E. M. Metabolic control mechanisms. 5. A solution for the equations representing interaction between

glycolysis and respiration in ascites tumor cells. *J. Biol. Chem.* 235, 2426–2439 (1960).

- Garfinkel, D. & Hess, B. Metabolic Control Mechanisms 7. A detailed computer model of the glycolytic pathway in ascites cells. *J. Biol. Chem.* 239, 971–983 (1964).
- Selkov, E. Self-Oscillations in Glycolysis 1. A Simple Kinetic Model. *Eur. J. Biochem.* 4, 79–86 (1968).
- Heinrich, R., Hermann-Georg, H. & Schuster, S. A theoretical approach to the evolution and structural design of enzymatic networks; Linear enzymatic chains, branched pathways and glycolysis of erythrocytes. *Bull. Math. Biol.* 49, 539–595 (1987)
- Heinrich, R., Montero, F., Klipp, E., Waddell, T. G. & Meléndez-Hevia, E. Theoretical approaches to the evolutionary optimization of glycolysis. Thermodynamic and kinetic constraints. *Eur. J. Biochem.* 243, 191–201 (1997).
- Teusink, B. *et al.* Can yeast glycolysis be understood terms of vitro kinetics of the constituent enzymes? Testing biochemistry. *Eur. J. Biochem.* **267**, 5313–5329 (2000).
- Smallbone, K., Simeonidis, E., Swainston, N. & Mendes, P. Towards a genome-scale kinetic model of cellular metabolism. *BMC Syst. Biol.* **4**, (2010).
- Stanford, N. J. *et al.* Systematic construction of kinetic models from genome-scale metabolic networks. *PLoS One* **8**, (2013).
- Saa, P. & Nielsen, L. K. A general framework for thermodynamically consistent parameterization and efficient sampling of enzymatic reactions. *PLoS Comput. Biol.* **11**, 1–25 (2015).
- Haiman, Z. B., Zielinski, D. C., Koike, Y., Yurkovich, J. T. & Palsson, B. O. MASSpy: Building, simulating, and visualizing dynamic biological models in Python using mass action kinetics. *PLoS Comput. Biol.* **17**, 1–20 (2021).
- John, P. C. S., Strutz, J., Broadbelt, L. J., Tyo, K. E. J. & Bomble, Y. J. Bayesian inference of metabolic kinetics from genome-scale multiomics data. *PLoS Comput. Biol.* **15**, 1–23 (2019).

The modified paragraph of the introduction is now as follows L23-33:

Living organisms constantly adapt to genetic or environmental perturbations. Describing these dynamic responses requires a deep understanding of the underlying network of biochemical and biophysical processes that comprise cellular metabolism. As cellular enzymes catalyze most metabolic processes, describing how perturbations propagate in large reaction networks requires knowledge and characterization of these enzymes' reaction mechanisms and kinetic properties. To this end, for over half a century, kinetic models of biochemical systems have been used to study a wide range of systems, from simple enzymatic reactions^{1,2} and small pathways³⁻⁶ to large-scale metabolic networks^{7,8} with diverse applications in health, biotechnology, systems and synthetic biology^{9,10}. Over the last years, more efforts have been made toward building genome-scale kinetic models^{11,12}, addressing and quantifying the uncertainty in the structure and parameters of kinetic models¹³⁻¹⁶, and identifying metabolic engineering design strategies using kinetic models¹⁷⁻¹⁹.

2.134, first sentence. This is an extreme understatement. Readers would be given a better idea of the scale of the problem by pointing out that even large databases of kinetic parameters such as BRENDA cannot furnish complete sets of parameters for central metabolism of any organism.

We thank the reviewer for this remark, and to address it, we modified the second paragraph of the introduction according to the reviewer's suggestion as following L34-38:

"Kinetic models employ a mathematical description of the enzymatic reaction rates, defined precisely as a function of the metabolite concentrations and kinetic parameters. However, experimental data detailing the parameters of an enzyme are scarce^{9,20}. Even large databases listing kinetic information, such as BRENDA²¹ and SABIO-RK²² do not comprise complete sets of parameters for the central metabolism of a single organism²⁰. ..."

3. I.194/5: "substrate product association dissociation"? Was "substrate enzyme" intended?

We thank the reviewer for this remark and corrected it as substrate association dissociation in the revised manuscript.

4. Fig. 3. Needs a magnification of about 2X to become decipherable, especially as there is a lot of fine detail to be absorbed. It would have helped if the legend had pointed out that the free energy pie charts had alternate segments representing the two branches of the mechanism rather than leaving the reader to decipher it.

We thank the reviewer for this remark, and we modified the Figure and the legend in the main text accordingly.

5. Ref. 21: "Hermann-Georg" is the Holzhütter of ref 25.

We thank the reviewer for this remark, and we corrected this error in the revised manuscript reference section.

6. Supplementary material has only two references, but the citation numbers are not always 1 and 2.

We thank the reviewer for this remark, and we corrected this error in the revised manuscript supplementary material.

Reviewer #2 (Remarks to the Author):

The authors present a framework to find optimal values for the kinetic constants in an enzymatic mechanism using MILP. This article follows in the tradition of the seminal works by Knowles and others and it is specially relevant now that optimization frameworks have proven to be valuable tools for analyzing other aspects of biochemistry and metabolism such as stoichiometry and thermodynamics.

The application of the framework produces results in agreement with the existing literature for simple mechanisms and enables the analysis of virtually any mechanism.

Right now, techniques like Flux Balance Analysis and its thermodynamical extensions enable an understanding of physiologically meaningful metabolic states in terms of flux distributions. Simultaneously, recent breakthroughs are showing the logic behind the allocation of resources in the cells with strategies such as proteome fractioning. The missing link between these two approaches is the catalytic efficiency of enzymes, a topic that is still poorly understood from a theoretical perspective. This article provides a valuable tool to help understand the evolution of enzymatic mechanisms.

The manuscript is well written, the methods are sound and the authors provide the code necessary for reproducing the results. I recommend accepting this manuscript as it is.

We thank the reviewer for the positive comments and encouraging words. We modified our Discussion (L390-396) in the revised manuscript to accentuate how our framework can be combined with genome-scale metabolic models and constraint-based modelling approaches to estimate a context-specific theoretical upper bound for the catalytic efficiency of metabolic enzymes.

Reviewer #3 (Remarks to the Author):

Sahin et al submitted the manuscript entitled "Optimal enzyme utilization suggests that concentrations and thermodynamics determine the operating binding mechanisms and enzyme saturations". The paper presents a theoretical work and an associated implemented MILP framework aiming at predicting the thermodynamics and kinetic parameters of Michaelis-Menten enzymatic systems. The originality of the work lies in the limited input data required for the prediction - elementary enzyme mechanisms, concentrations of substrates and products, thermodynamics properties - and the application to a random-ordered multi-substrate system. The manuscript is very well written, and demonstrates the quality of the work. The code associated to the manuscript works and is well documented.

We thank the reviewer for the encouraging and positive words.

My comments are the following:

- How could this theoretical work be applied in practice to networks of biochemical reactions? If one wants to analyse a full metabolic pathway (several enzymes working as a cascade) or a larger network, is it possible and how would they proceed?

We thank the reviewer for this question and for opening an exciting discussion. Our framework can easily be applied to networks of biochemical reactions to determine optimal modes of operations for a cascade of reactions or a whole metabolic pathway. We are currently working on a continuation project where we apply our framework to metabolic pathways and networks. We briefly discussed the potential applications below.

The only connections between different enzymes working in a cascade are through the steady-state fluxes, which define a system of linear equations, and through the thermodynamic displacement of each reaction from equilibrium which is a function of the standard Gibbs free energy ($\Delta_r G'^{\circ}$) and metabolite concentrations. Therefore, our framework can be directly applied to study a full metabolic pathway after having the necessary inputs to the framework, which are the metabolite concentrations, thermodynamic properties, and the reaction mechanism. With the recent advances in the genome-scale metabolic model reconstructions and their analysis with constraint-based modeling methods such as Thermodynamics-based Flux Analysis (TFA)^{23,24} we can integrate quantitative metabolomics data and estimate thermodynamically feasible flux and concentration profiles as well as how far each reaction operates from thermodynamic equilibrium. The only remaining input then is the reaction mechanisms, which are assigned based on the molecularity and previous knowledge of the mechanistic regulatory details of each reaction⁹. As our framework maximizes the net steady-state flux over the enzyme concentration for each reaction, we can run our framework independently for each reaction in the network after having the required inputs. Then, using the steady-state flux distributions from TFA analysis, we can estimate the enzyme abundance for each reaction to match the catalytically optimal enzyme kinetics. Therefore, our framework scales linearly with the increasing number of reactions working in a cascade, as it can be performed independently for each reaction.

Another application of our framework to metabolic networks will be to compare our estimates at a catalytically optimal state with the *in vivo* and *in vitro* catalytic rates. In essence, what our

framework optimizes for is equivalent to the "apparent catalytic rate" (k_{app}), which was first introduced by Valgepea et al.²⁵ and has been recently used in the context of genome-scale metabolic models to estimate the catalytic rate of enzymes *in vivo*^{26,27} using enzyme abundance data and flux estimates. Our framework can also be used in this context to provide a theoretical upper bound for the apparent catalytic rate (k_{app}^{opt}) and to assess how far the enzymes *in vivo* operate from their theoretical optimum.

-An added value of the work is the limited necessary inputs. How easy are they to obtain in practice and how does it prevent the scalability of the predictions to large(r) enzymatic systems? You mention in the introduction the difficulty for a cell containing thousands of enzymatic reactions, but not how your approach actually facilitates the task.

We thank the reviewer for this question, and we believe that this question relates to the previous one from the reviewer. We rephrased some of our answers above with more focus on the required inputs; for the remaining questions, we refer the reviewer to the previous answer.

The necessary inputs for our framework to estimate modes of operations of an optimally utilized enzyme are the reaction mechanism, metabolite concentrations, and thermodynamic properties in terms of standard Gibbs free energy of the reaction. The reaction mechanism can be assigned based on the molecularity of the reaction or on previous knowledge and literature⁹. The second input is the metabolite concentrations which are partially available in the literature for well-studied organisms^{28,29}. Moreover, constraint-based modelling frameworks like TFA²³ can integrate such data and estimate thermodynamically feasible concentration profiles for all the metabolites in the biochemical network. The last input is the reaction's thermodynamic properties, which can be estimated based on group-contribution³⁰ or component-contribution³¹ methods. After obtaining the necessary inputs, our framework scales linearly with the number of reactions, as each reaction can be studied independently. (Previous question)

As our framework scales linearly and provides a systematic approach to estimate optimal modes of operations for any reaction mechanism, it can be easily applied to larger systems, such as a network of reactions or an entire metabolic pathway. Furthermore, as we formulate the problem as a MILP, our framework can also be used to estimate kinetic parameters that are consistent with the steady-state flux physiology and thus can be combined with kinetic modelling toolboxes to capture the dynamic behaviour of metabolism.

Following the previous two questions, we modified the Discussion in the revised manuscript as follows L379-396:

“An added value of our framework is the limited necessary inputs which are the reaction mechanism, thermodynamic properties of the reaction, and the metabolite concentrations. The reaction mechanism can be assigned based on the molecularity of the reaction or on previous knowledge and literature⁹. The standard Gibbs free energies of the reactions can be estimated by group-contribution³⁰ or component-contribution³¹ methods. Lastly, the metabolite concentrations can be obtained partially from the literature for well-studied organisms^{28,29,32}, or can be estimated for an entire metabolic pathway using constraint-based optimization methods such as Thermodynamics-based Flux Analysis (TFA)^{23,24}. Using TFA^{23,24}, we can integrate quantitative metabolomics and fluxomics data and estimate thermodynamically feasible concentration and flux profiles as well as how far each reaction operates from thermodynamic equilibrium by sampling the thermodynamically feasible concentration and flux profile space, as it is done in the construction of population of kinetics models¹⁴. After having the necessary inputs, our framework can be easily applied to study a network of enzymatic reactions or an entire metabolic pathway under the

assumption that each reaction operates at a catalytically optimal state. This procedure scales linearly with the number of reactions in the network. Thus, by combining experimental measurements with genome-scale metabolic models and constraint-based modeling approaches, our framework can estimate a context-specific theoretical upper bound for the catalytic efficiency of metabolic enzymes with detailed kinetic and thermodynamic considerations as they operate in vivo.

Could you provide validation of your optimisation approach on existing experimental data?

We thank the reviewer for this question. Our approach could be used to assess how far an enzyme (*in vitro* or *in vivo*) operates from its optimally utilized counterpart, and it can provide a comparison rather than validation in conventional ways with experimental data. A similar analysis was done previously by Klipp and Heinrich³³ where they applied their theoretical model to interpret the kinetic data of triosephosphate isomerase based on the elementary rate constants measured by Knowles and Albery³⁴.

We avoided to compare with the values reported in the databases, because we believe that if these reported values are the result of *in vivo* optimization of enzymes, we will also need to use experimentally reference conditions and the network framework approach we mentioned earlier. We are currently working on a project where using our framework; we estimate how far an enzyme operates from its catalytically optimal state, which is beyond the scope of this work. Such comparison with existing experimental data will be also based on macroscopic kinetic parameters, which are relatively easier to measure experimentally. Firstly, the *in vitro* measured quantities, such as catalytic rate constants, k_{cat} 's and Michaelis constants, K_M 's can be compared with estimations from our framework for a catalytically optimal enzyme. Secondly, for a physiologically more relevant comparison, our framework can also be used against high throughput *in vivo* measurements. As mentioned in the previous responses, using quantitative proteomics and flux data we can calculate the *in vivo* apparent rate constant (k_{app}) and use it to characterize the catalytic efficiency of the enzyme *in vivo*. Comparing the apparent rate constant with the output of our framework (k_{app}^{opt}), we can have an estimation of the extent of catalytic efficiency of an enzyme. (Please see also below on the discussion about L344, on enzymes' moderate catalytic efficiency. As previous researchers suggested, we also agree that there many physiological reasons why enzymes in various pathways will operate below catalytic efficiency. Therefore, it will be difficult to draw any conclusion by comparing the optimal parameter values we have estimated with the reported *in vitro* values).

Results L216-224: is all the paragraph about FigS6? If yes, maybe refer to it earlier (eg L217-219)

We thank the reviewer for the feedback. It is modified in the revised manuscript as suggested.

Results on multi-substrate enzymes.

- I could not relate the observations to the figure at L246-250

We thank the reviewer for pointing out this issue, and we apologize for the lack of clarity. We modified the L271-280 in the original manuscript as follows:

"A variability analysis revealed that this flexibility had a range around $\alpha=0.5$. This led to alternative values for the elementary rate constants and, consequently to alternative modes of net fluxes through the upper and lower branches. More interestingly, when the substrate concentrations are equal and less than product concentrations, there exists a unique configuration of elementary constants for the optimally utilized enzymes; instead, when they are comparable or slightly higher than product

concentrations, there can exist alternative values for the elementary constants that can result in optimal enzyme utilization. This suggests that the product concentration relative to the concentration of the substrates affects the uniqueness of the optimal solution and, therefore, the preferential binding for the unbiased Bi-Uni mechanism."

- L273-274, can you refer to the blue and red triangles of Fig 3a and white triangle of fig 3a?

We modified the main text accordingly to refer to the selected operating conditions (L302-305):

Similar to the reversible Michaelis-Menten mechanism, we analysed three prototypical operating conditions: two symmetric operating conditions for distinct substrate concentrations (red and blue triangles in Figure 3a) and one with identical concentrations for the substrates (white dashed square in Figure 3a).

- L285 "...binding of substrate B": maybe add "(steps 2 or 5)"?

We thank the reviewer for the clarification. We included this in the revised version of the manuscript (L316)

Discussion L344: any possible explanation on why enzymes do not operate at optimal efficiency and how your model could help understand the reasons?

We thank the reviewer for this question. The possible reasons for enzymes' moderate catalytic efficiency have been discussed in detail by Ron Milo and coworkers^{35,36}. Although a detailed review of this is beyond the scope of this work, we briefly summarized some of their findings below. We also explained how our framework could help understand the complex interplay between physicochemical constraints and evolutionary selective pressure on enzyme kinetics.

Two possible factors have been proposed to explain the non-optimal catalytic efficiency of enzymes which are the evolutionary pressure and physicochemical constraints^{35,36}. However, the effects of these factors on enzyme kinetic parameters are still unclear. Firstly from an evolutionary perspective, the extent of the evolutionary pressure could vary in different metabolic pathways. This is hypothesized to be proportional to the contribution of these pathways to the organism's fitness³⁵. Moreover, the catalytic efficiency is not the sole determinant of the evolutionary pressure; hence moderate catalytic efficiency might also be an outcome of the trade-off between different evolutionary pressures³⁶⁻³⁸ such as metabolite loads, selectivity, and regulation. Secondly, considering physicochemical constraints and physiological demands, we cannot generalize a unique optimal state that applies to all enzymes. The catalytic efficiency and its theoretical limit are indeed reaction, mechanism, and context (metabolite concentrations, thermodynamics) dependent³⁶. For example, it has been previously suggested that the enzymes operating close to thermodynamic equilibrium represent flux bottlenecks and are under higher evolutionary pressure for increased catalytic efficiency. We should also keep in mind that most of the kinetic parameters used to evaluate the catalytic efficiency of enzymes are measured under *in vitro* conditions which do not represent well the *in vivo* physicochemical constraints.

Our framework can shed light on the driving forces and constraints that have shaped moderately efficient enzymes in three ways. First of all, using our framework, we can estimate a condition-specific (operating conditions, mechanism) theoretical upper bound for the catalytic efficiency of any enzyme mechanism, which will provide a more accurate comparison with the natural enzymes to assess how far they operate from their theoretical optimum. Secondly, as we formulate the problem as a MILP, we can explore the suboptimal solutions

using traditional sampling methods and explore the fitness landscape of enzymes toward catalytic perfection, which was not possible with previous methodologies. This way, we can understand the independent contribution of each variable to the fitness landscape of the moderately efficient enzymes and have a detailed understanding of how diminishing returns and trade-offs affect the evolutionary trajectory of enzymes toward optimal catalytic efficiency. Lastly, we can integrate alternative objectives by including them in the objective function to study different evolutionary pressures and trade-offs between them.

Following this, we have modified the discussion L397-413 as follows:

"In this study, our analysis was focused on and limited to the kinetic design and modes of operations of enzymes at maximal reaction rates, though our computational framework would also allow the exploration of alternative and suboptimal solutions. Therefore, although we are still far from completely understanding the complex interplay between the physicochemical constraints and evolutionary pressures that shape enzyme catalysis, our framework can help us map the parametric domain for a broad range of operating conditions and shed light on the driving forces and constraints that have shaped natural enzymes. First of all, using our framework, we can estimate a condition-specific theoretical upper bound for the catalytic efficiency of any enzyme mechanism, which will provide a more accurate comparison with the natural enzymes to assess how far they operate from their theoretical optimum. Secondly, as we formulate the problem as a MILP, we can explore the suboptimal solutions using traditional sampling methods and explore the fitness landscape of enzymes toward catalytic perfection, which was not possible with previous methodologies. This way, we can understand the independent contribution of each variable to the fitness landscape of the moderately efficient enzymes and have a detailed understanding of how diminishing returns and trade-offs affect the evolutionary trajectory of enzymes toward optimal catalytic efficiency. Lastly, we can integrate alternative objectives by including them in the objective function to study different evolutionary pressures and trade-offs between them"

Figure 1. The figure is great but it has a lot of contents associated in the main text and it is hard to see which part of the figure the text refers to. Could you refer to the specific subparts of the figure in the text (e.g. Figure 1 panel 1) rather than just adding "Figure 1"?

We thank the reviewer for the positive words and for the suggestion. We modified the main text in the revised manuscript to refer to specific subparts of Figure 1.

Figure 2. Panel a. Could you add information about the thermodynamic variable Γ in the legend? And mention as well the dashed line for $\Gamma = 1$?

We thank the reviewer for this comment. We modified the legend of the figure to include this information.

Figure 3.

- Panel d has a lot of content and is referred to quite a few times in the text. Would it make sense to split d into d and e, d being the part of the blue and red triangles?
- The text in the pie plots is a bit small
- The legend could be extended a bit: what do the dotted lines in the pie charts mean? Why are the right pie charts combined to a sub-pie chart?

- The 3 alpha values in the white triangles (0.35 to 0.65) are unclear at first sight as they correspond to a single triangle in 3a. Maybe emphasize on the meaning in the legend of lower d), or instead you could choose a color different from white in a) for those points (because white is 0.5).

We thank the reviewer for his suggestions in Figure 3. We reorganized and modified Figure 3 as well as its legend accordingly in the revised main text.

Supplementary figures:

- the right subpanels in several supp figs are not selfcontained as the y-axis label is only displayed on the left subpanel: could you add the label to the other plots?

We modified the figures in the revised supplementary material.

- Fig S2. A dashed line is mentioned in Fig S2's legend but is not present in the plot.

We thank the reviewer for this remark. It is corrected in the revised supplementary material.

- Fig S3. I couldn't find any reference to it in the text.

We thank the reviewer for this remark, and we added the reference in the revised manuscript.

L 186-188:

"We further calculated forward and backward turnover numbers at optimal state and showed their sensitivity over the concentration space (Supplementary Fig. 3)."

Ref 34 of main text and ref 2 of supp show a switch between the first name and name of the first author.

We thank the reviewer for this remark, and we corrected it in the revised version of the main and supplementary text.

Additional changes to the manuscript

- Reformulated the introduction to include more details on the previous works on enzyme optimality.
- Added a Supplementary Note named "Discussion and comparison of the existing mathematical methodologies on enzyme optimality" to include an in-depth discussion and comparison of the existing mathematical methodologies.

References

1. Selkov, E. Self-Oscillations in Glycolysis 1. A Simple Kinetic Model. *Eur. J. Biochem.* **4**, 79–86 (1968).
2. Heinrich, R., Holzhütter, H. G. & Schuster, S. A theoretical approach to the evolution and structural design of enzymatic networks; Linear enzymatic chains, branched pathways and glycolysis of erythrocytes. *Bull. Math. Biol.* **49**, 539–595 (1987).
3. Chance, B., Garfinkel, D., Higgins, J., Hess, B. & Chance, E. M. Metabolic control mechanisms. 5. A solution for the equations representing interaction between glycolysis and respiration in ascites tumor cells. *J. Biol. Chem.* **235**, 2426–2439 (1960).
4. Garfinkel, D. & Hess, B. Metabolic Control Mechanisms 7. A detailed computer model of the glycolytic pathway in ascites cells. *J. Biol. Chem.* **239**, 971–983 (1964).
5. Heinrich, R., Montero, F., Klipp, E., Waddell, T. G. & Melendez-Hevia, E. *Theoretical*

- approaches to the evolutionary optimization of glycolysis Chemical analysis. Eur. J. Biochem* vol. 244 (1997).
6. Teusink, B. *et al.* Can yeast glycolysis be understood terms of vitro kinetics of the constituent enzymes? Testing biochemistry. *Eur. J. Biochem.* **267**, 5313–5329 (2000).
 7. Stanford, N. J. *et al.* Systematic construction of kinetic models from genome-scale metabolic networks. *PLoS One* **8**, (2013).
 8. Chakrabarti, A., Miskovic, L., Soh, K. C. & Hatzimanikatis, V. Towards kinetic modeling of genome-scale metabolic networks without sacrificing stoichiometric, thermodynamic and physiological constraints. *Biotechnol. J.* **8**, 1043–1057 (2013).
 9. Miskovic, L., Tokic, M., Fengos, G. & Hatzimanikatis, V. Rites of passage: Requirements and standards for building kinetic models of metabolic phenotypes. *Curr. Opin. Biotechnol.* **36**, 146–153 (2015).
 10. Islam, M. M., Schroeder, W. L. & Saha, R. Kinetic modeling of metabolism: Present and future. *Curr. Opin. Syst. Biol.* **26**, 72–78 (2021).
 11. Smallbone, K., Simeonidis, E., Swainston, N. & Mendes, P. Towards a genome-scale kinetic model of cellular metabolism. *BMC Syst. Biol.* **4**, (2010).
 12. Khodayari, A., Zomorodi, A. R., Liao, J. C. & Maranas, C. D. A kinetic model of Escherichia coli core metabolism satisfying multiple sets of mutant flux data. *Metab. Eng.* **25**, 50–62 (2014).
 13. Haiman, Z. B., Zielinski, D. C., Koike, Y., Yurkovich, J. T. & Palsson, B. O. MASSpy: Building, simulating, and visualizing dynamic biological models in Python using mass action kinetics. *PLoS Comput. Biol.* **17**, 1–20 (2021).
 14. Miskovic, L. & Hatzimanikatis, V. Production of biofuels and biochemicals: In need of an ORACLE. *Trends Biotechnol.* **28**, 391–397 (2010).
 15. Tran, L. M., Rizk, M. L. & Liao, J. C. Ensemble modeling of metabolic networks. *Biophys. J.* **95**, 5606–5617 (2008).
 16. Saa, P. & Nielsen, L. K. A general framework for thermodynamically consistent parameterization and efficient sampling of enzymatic reactions. *PLoS Comput. Biol.* **11**, 1–25 (2015).
 17. Tokic, M., Hatzimanikatis, V. & Miskovic, L. Large-scale kinetic metabolic models of Pseudomonas putida KT2440 for consistent design of metabolic engineering strategies. *Biotechnol. Biofuels* **13**, 1–19 (2020).
 18. Androozzi, S., Miskovic, L. & Hatzimanikatis, V. ISCHRUNK - In Silico Approach to Characterization and Reduction of Uncertainty in the Kinetic Models of Genome-scale Metabolic Networks. *Metab. Eng.* **33**, 158–168 (2016).
 19. John, P. C. S., Strutz, J., Broadbelt, L. J., Tyo, K. E. J. & Bomble, Y. J. Bayesian inference of metabolic kinetics from genome-scale multiomics data. *PLoS Comput. Biol.* **15**, 1–23 (2019).
 20. Foster, C. J., Wang, L., Dinh, H. V., Suthers, P. F. & Maranas, C. D. Building kinetic models for metabolic engineering. *Curr. Opin. Biotechnol.* **67**, 35–41 (2021).
 21. Chang, A. *et al.* BRENDA, the ELIXIR core data resource in 2021: New developments and updates. *Nucleic Acids Res.* **49**, D498–D508 (2021).
 22. Wittig, U., Rey, M., Weidemann, A., Kania, R. & Müller, W. SABIO-RK: An updated

- resource for manually curated biochemical reaction kinetics. *Nucleic Acids Res.* **46**, D656–D660 (2018).
23. Salvy, P. *et al.* PyTFA and matTFA: A Python package and a Matlab toolbox for Thermodynamics-based Flux Analysis. *Bioinformatics* **35**, 167–169 (2018).
 24. Henry, C. S., Broadbelt, L. J. & Hatzimanikatis, V. Thermodynamics-based metabolic flux analysis. *Biophys. J.* **92**, 1792–1805 (2007).
 25. Valgepea, K., Adamberg, K., Seiman, A. & Vilu, R. Escherichia coli achieves faster growth by increasing catalytic and translation rates of proteins. *Mol. Biosyst.* **9**, 2344–2358 (2013).
 26. Davidia, D. *et al.* Global characterization of in vivo enzyme catalytic rates and their correspondence to in vitro kcat measurements. *Proc. Natl. Acad. Sci. U. S. A.* **113**, 3401–3406 (2016).
 27. Heckmann, D. *et al.* Machine learning applied to enzyme turnover numbers reveals protein structural correlates and improves metabolic models. *Nat. Commun.* **9**, (2018).
 28. Park, J. O. *et al.* Metabolite concentrations, fluxes and free energies imply efficient enzyme usage. *Nat. Chem. Biol.* **12**, 482–489 (2016).
 29. Bennett, B. D. *et al.* Absolute metabolite concentrations and implied enzyme active site occupancy in Escherichia coli. *Nat. Chem. Biol.* **5**, 593–599 (2009).
 30. Jankowski, M. D., Henry, C. S., Broadbelt, L. J. & Hatzimanikatis, V. Group contribution method for thermodynamic analysis of complex metabolic networks. *Biophys. J.* **95**, 1487–1499 (2008).
 31. Beber, M. E. *et al.* EQuilibrator 3.0: A database solution for thermodynamic constant estimation. *Nucleic Acids Res.* **50**, D603–D609 (2022).
 32. McCloskey, D. *et al.* A model-driven quantitative metabolomics analysis of aerobic and anaerobic metabolism in E. coli K-12 MG1655 that is biochemically and thermodynamically consistent. *Biotechnol. Bioeng.* **111**, 803–815 (2014).
 33. Klipp, E. & Heinrich, R. Evolutionary optimization of enzyme kinetic parameters; effect of constraints. *J. Theor. Biol.* **171**, 309–323 (1994).
 34. Knowles, J. R. & Albery, W. J. Perfection in Enzyme Catalysis: The Energetics of Triosephosphate Isomerase. *Acc. Chem. Res.* **10**, (1977).
 35. Bar-Even, A. *et al.* The moderately efficient enzyme: Evolutionary and physicochemical trends shaping enzyme parameters. *Biochemistry* **50**, 4402–4410 (2011).
 36. Davidi, D., Longo, L. M., Jabłońska, J., Milo, R. & Tawfik, D. S. A Bird’s-Eye View of Enzyme Evolution: Chemical, Physicochemical, and Physiological Considerations. *Chem. Rev.* **118**, 8786–8797 (2018).
 37. Newton, M. S., Arcus, V. L., Gerth, M. L. & Patrick, W. M. Enzyme evolution: innovation is easy, optimization is complicated. *Curr. Opin. Struct. Biol.* **48**, 110–116 (2018).
 38. Tepper, N. *et al.* Steady-State Metabolite Concentrations Reflect a Balance between Maximizing Enzyme Efficiency and Minimizing Total Metabolite Load. *PLoS One* **8**, 1–13 (2013).

REVIEWERS' COMMENTS

Reviewer #3 (Remarks to the Author):

Thanks to the authors for addressing the comments.

I have no further remark and I recommend the acceptance of the manuscript.

Authors' response to the reviewers' comments

Title: Optimal enzyme utilization suggests that concentrations and thermodynamics determine binding mechanisms and enzyme saturations

Manuscript ID: NCOMMS-22-42599A

Authors: Sahin A., Weilandt D. R., and Hatzimanikatis V.

Reviewer comments

Reviewer #3 (Remarks to the Author):

Thanks to the authors for addressing the comments.

I have no further remark and I recommend the acceptance of the manuscript.

We are glad that the revised manuscript has successfully addressed the reviewer's comments. We thank the reviewer for all the insightful comments and suggestions.